# Fault Diagnosis of Rolling Bearings Based on HFMD and Dual-Branch Parallel Network Under Acoustic Signals

**DOI:** 10.3390/s25175338

**Published:** 2025-08-28

**Authors:** Hengdi Wang, Haokui Wang, Jizhan Xie

**Affiliations:** School of Mechanical and Electrical Engineering, Henan University of Science and Technology, Luoyang 471023, China; hk13526970213@163.com (H.W.); 13148054890@163.com (J.X.)

**Keywords:** acoustic signal, dual-branch parallel network, eagle fish optimization algorithm, feature mode decomposition, rolling bearing, fault diagnosis

## Abstract

This paper proposes a rolling bearing fault diagnosis method based on HFMD and a dual-branch parallel network, aiming to address the issue of diagnostic accuracy being compromised by the disparity in data quality across different source domains due to sparse feature separation in rolling bearing acoustic signals. Traditional methods face challenges in feature extraction, sensitivity to noise, and difficulties in handling coupled multi-fault conditions in rolling bearing fault diagnosis. To overcome these challenges, this study first employs the HawkFish Optimization Algorithm to optimize Feature Mode Decomposition (HFMD) parameters, thereby improving modal decomposition accuracy. The optimal modal components are selected based on the minimum Residual Energy Index (REI) criterion, with their time-domain graphs and Continuous Wavelet Transform (CWT) time-frequency diagrams extracted as network inputs. Then, a dual-branch parallel network model is constructed, where the multi-scale residual structure (Res2Net) incorporating the Efficient Channel Attention (ECA) mechanism serves as the temporal branch to extract key features and suppress noise interference, while the Swin Transformer integrating multi-stage cross-scale attention (MSCSA) acts as the time-frequency branch to break through local perception bottlenecks and enhance classification performance under limited resources. Finally, the time-domain graphs and time-frequency graphs are, respectively, input into Res2Net and Swin Transformer, and the features from both branches are fused through a fully connected layer to obtain comprehensive fault diagnosis results. The research results demonstrate that the proposed method achieves 100% accuracy in open-source datasets. In the experimental data, the diagnostic accuracy of this study demonstrates significant advantages over other diagnostic models, achieving an accuracy rate of 98.5%. Under few-shot conditions, this study maintains an accuracy rate no lower than 95%, with only a 2.34% variation in accuracy. HFMD and the dual-branch parallel network exhibit remarkable stability and superiority in the field of rolling bearing fault diagnosis.

## 1. Introduction

Rolling bearings serve as critical components in rotating machinery, significantly influencing the stability and safety of mechanical systems. Due to prolonged operation, bearings are susceptible to failure, which can result in diminished equipment performance and may even lead to severe accidents [1,2,3]. The monitoring and diagnosis of bearing faults are crucial for enhancing the reliability and productivity of mechanical systems. Compared to traditional contact-based vibration measurement methods, bearing fault diagnosis techniques that utilize acoustic signals present several advantages, including non-contact operation, ease of deployment, and high adaptability, making them particularly suitable for challenging environments such as high temperatures and corrosive conditions. Acoustic signals provide comprehensive information regarding the operational status of bearings; however, they are significantly influenced by environmental noise. Therefore, the development of efficient noise reduction methods and feature extraction techniques has emerged as a key focus of research [4,5,6,7].

Methods based on physical models possess high accuracy and interpretability; however, establishing models under practical conditions is challenging. Signal processing techniques can effectively extract features, but their capability to handle non-stationary signals is limited. Data-driven methods excel in managing complex patterns and automatic feature extraction; however, their primary obstacles are the reliance on large amounts of labeled data and the high computational cost of model training [8,9]. This study focuses on integrating the advantages of multiple methods to enhance the intelligence and adaptability of diagnostics.

The feature extraction methods for acoustic signals primarily include analyses in the time domain, frequency domain, and time-frequency domain. These methods are capable of extracting critical fault information from the original signals, characterizing the operational status of the equipment, and facilitating fault diagnosis [10]. Signal decomposition technology is extensively utilized in fault diagnosis, as it effectively disaggregates complex acoustic signals into multiple modal components, eliminates noise, and extracts fault characteristic information. By integrating signal decomposition with time-frequency analysis methods, the accuracy of fault pattern recognition and the robustness of the diagnostic system can be significantly enhanced [11]. Empirical Mode Decomposition (EMD) [12] can adaptively decompose nonlinear and non-stationary signals into several Intrinsic Mode Functions (IMFs), thus facilitating the extraction of local fault characteristics. However, it is susceptible to endpoint effects and mode mixing, which can result in significant energy leakage between components. The improved Complete Ensemble Empirical Mode Decomposition (CEEMD) [13] enhances the noise control strategy based on EEMD, further suppressing mode mixing and reducing residual noise; however, the results are more sensitive to the amplitude and combination of noise. Variational Mode Decomposition (VMD) [14] utilizes a variational optimization framework along with adaptive bandpass filtering to enhance boundary effect suppression and improve decomposition stability. However, it necessitates pre-specifying the number of modes and the penalty parameter, rendering it sensitive to parameter selection and resulting in higher experimental tuning costs. To address the challenges associated with VMD parameter dependency and IMF selection, Zhang et al. [15] introduced the Grasshopper Optimization Algorithm (GOA). This algorithm autonomously searches for the optimal number of modes and bandwidth control parameters, utilizing the weighted kurtosis index as the objective function, and extracts fault features through kurtosis-sensitive mode extraction. Li et al. [16] combined maximum envelope kurtosis with frequency band entropy (FBE) to adaptively determine the number of modes and select fault information IMFs (IMFs). Although EMD and VMD can enhance feature expression, they have limitations in terms of generalization.

In the field of rotating machinery fault diagnosis, Feature Mode Decomposition (FMD) [17] is recognized as an effective signal analysis method. The advantages of FMD include its excellent noise resistance and adaptability to fault cycles, which enable effective feature extraction without requiring prior knowledge of the fault period. However, the performance of FMD is sensitive to parameters such as filter length and the number of decomposition modes; inappropriate parameter selection can result in suboptimal decomposition outcomes. To address this challenge, Jia [18] employed the Whale Optimization Algorithm (WOA) in conjunction with envelope entropy as the fitness function to optimize the parameters of FMD, thereby enhancing its ability to extract fault features in noisy environments. Sumika [19] employed the Artificial Hummingbird Algorithm (AHA) to optimize the parameters of FMD and introduced the Sparse Impact Measurement Index as a health indicator, thereby enhancing the accuracy of FMD in extracting bearing fault characteristics. Yan [20] integrated the Particle Swarm Optimization (PSO) algorithm with the Signal Cycle Kurtosis Noise Ratio (SCKNR) index to propose a parameter-optimized FMD method. This integration enhances the efficiency of fault feature extraction while reducing reliance on manual expertise.

In recent years, the application of deep learning methods in bearing fault diagnosis has gradually shifted from model stacking and accuracy improvement to deeper research dimensions, such as feature interpretability, physics-guided approaches, and multi-source fusion. Scholars have proposed distinctive diagnostic strategies from various perspectives. Feng et al. [21] proposed a Bayesian deep learning method based on Physics-Informed Neural Networks (PINN), which considers the uncertainty of material manufacturing process parameters. Under a given stress ratio, this method establishes a quantitative relationship between the microstructural features of materials and their fatigue performance using Bayesian deep learning techniques. Yan et al. [22] investigated a multi-modal imitation learning-based arc detection network (MILADNet) for arc detection in the pantograph-contact wire system within complex railway environments. By integrating features from infrared and visible light images and introducing an online imitation learning framework along with an unsupervised transferable representation learning method, the model’s detection performance was enhanced under challenging conditions such as strong reflections and adverse weather. Furthermore, Yan et al. [23] developed an unsupervised machine anomaly sound detection model (Unsuper-TDGCN) based on Transformers and Dynamic Graph Convolutional Networks (DyGCN), aiming to address the domain shift issue caused by variations in acoustic features of the same equipment under different operating conditions, and validated its effectiveness across multiple datasets.

Bai et al. [24] proposed a Target-Specific Weight-Based Transfer Learning Adversarial Network (TSWN) that enhances the performance of target domain classifiers by facilitating the transfer of shared features between the source and target domains through a weight learning strategy and adversarial training. However, its performance may significantly decline in scenarios with substantial label space differences, limiting its applicability. Wang et al. [25] effectively extracted fault features by transforming one-dimensional vibration signals into two-dimensional time-frequency maps through the combination of wavelet transform and deep transfer learning. Dai et al. [26] optimized the VMD parameters using PSO to enhance the signal decomposition effect, selected signal components using the Weighted Composite Kurtosis (WCK) index and achieved fault identification by integrating GADF and Swin Transformer. However, the performance on small sample datasets has not been thoroughly validated. Ruan et al. [27] proposed a physically guided convolutional neural network (PGCNN) that leverages the physical attributes of signals. The model design incorporates the periodicity of faults and the characteristics of envelope attenuation. Additionally, it adopts a rectangular convolutional kernel structure in place of the traditional square kernel, achieving a joint optimization of convolutional parameters and input forms. Zhang et al. [28] proposed the Hybrid Attention-Improved Residual Network (HA-ResNet), which integrates wavelet packet band selection with a channel attention mechanism, thereby effectively enhancing the expressive power of time-frequency features. Yu et al. [29] constructed a graph structure using CNN feature maps and introduced a node voting strategy along with an edge dropping mechanism to enhance the model’s adaptability to varying working conditions, thereby achieving stable diagnosis across different operational scenarios. To address this limitation, Yin [30] proposed an improved DRSN-GRU dual-channel model, which enhances noise processing and compound fault recognition capabilities. However, this model primarily targets one-dimensional data processing and does not sufficiently exploit the multi-dimensional characteristics of the data. Hu [31] introduced a bearing fault diagnosis method that utilizes Markov Transition Fields, Continuous Wavelet Transform (CWT), and a dual-channel CNN. Nevertheless, it has a significant drawback: it requires a substantial volume of data. When the data volume is limited, the fault information is restricted, making it challenging to achieve high accuracy.

This paper proposes a rolling bearing fault diagnosis method that utilizes the HawkFish Optimization Algorithm to optimize FMD (HFMD) and employs a dual-branch parallel network. Firstly, the HFMD decomposition algorithm is employed to process the original sound signals, effectively mitigating the issue of information aliasing caused by the sensitivity of modal parameters and structural disturbances. Subsequently, the optimal mode containing fault information is selected as input and injected into two feature extraction pathways: on one hand, the time-domain channel introduces a combination of the Res2Net multi-scale residual structure and the Efficient Channel Attention (ECA) module to enhance the resolution and expressive power of local patterns; on the other hand, the time-frequency channel constructs a time-frequency map based on CWT and integrates a multi-stage cross-scale attention mechanism with Swin Transformer to model global dependencies across frequency bands. The Multi-Scale Cross-Scale Attention (MSCSA) achieves information fusion of channel space across multiple heads and layers. The ECA module highlights key responses and suppresses redundant noise through a channel re-weighting mechanism. By leveraging a collaborative learning mechanism and structural preservation constraints, the dual-pathway realizes consistent alignment of cross-modal features, comprehensively mining global time-frequency relationships and local temporal details, thereby enhancing the perception efficiency and diagnostic robustness of fault recognition. The dual-branch parallel network synchronously extracts fault features at different scales and interacts at the pixel or channel level in the fusion layer, significantly improving the multi-scale feature capture capability and robustness in complex noise environments. By capitalizing on the complementary information flow between branches, it enhances the model’s interpretability and generalization performance. The innovation of this model lies in two key aspects:

(1) By aggregating features of different resolutions from Stage 1 to Stage 4 through MSCSA, it utilizes cross-scale attention to efficiently facilitate cross-scale information interaction within local windows while maintaining the relative independence of multi-stage information in Intra-FFN. This approach enhances the recognition of time-frequency components of weak fault signals.

(2) The Swin Transformer constructs a hierarchical structure of shifted windows, ensuring linear complexity while balancing local and global aspects.

(3) Res2Net refines local features and enlarges the equivalent receptive field by employing parallel multi-scale subdivision within a single residual block, and ECA significantly enhances the feature recalibration effect across branches by modeling inter-branch interactions through adaptive 1DConv.

## 2. Methodologies

### 2.1. Feature Mode Decomposition

FMD, proposed by MIAO, is an innovative signal processing method utilized for the diagnosis of faults in rotating machinery [17]. Its mechanism is realized through an adaptive finite impulse response (FIR) filter bank, with correlated kurtosis (*CK*) serving as the optimization objective function. This approach simultaneously captures the transient impact characteristics and periodic modulation patterns of the signal. The FMD method precisely extracts the fault characteristic frequency by alternately updating the filter parameters and period estimates, effectively suppressing mode mixing and the generation of spurious components. By comprehensively considering the impulsiveness and periodicity of the signal, the algorithm can effectively resist the influence of various interferences and noise. The flowchart of the FMD decomposition process is illustrated in Figure 1, The *CK* representation of the *k*-th decomposition function *u_k_* is as follows.(1)$$CK(u_{k})=\frac{\sum_{n=1}^{N}{\left(\sum_{m=0}^{M}u_{k}\left(n-mT_{s}\right)\right)}^{2}}{\sum_{n=1}^{N}u_{k}{\left(n\right)}^{2}}$$

In the equation, ***N*** is the total number of sampling points for the decomposition function *u_k_*; *n* is the index of the current sampling point; *M* is the maximum delay when calculating the related kurtosis; *m* is the index of the related delay; *T_s_* is the sampling time interval; and *u_k_*(*n*) is the amplitude of the *k*-th decomposition function at the *n*-th sampling point in the time series.

The representation for updating the coefficients *f_k_* of the *k*-th FIR filter is as follows:(2)$$R_{XWX}f_{k}=R_{XX}f_{k}\lambda$$

In the equation ***R****_XWX_* is the weighted correlation matrix, which enhances the significance of frequency components by considering the statistical properties of the signal; ***R****_XX_* is the autocorrelation matrix, which measures the similarity of the signal with itself at different time delays; *f_k_* is the coefficient vector of the *k*-th FIR filter; and *λ* is the eigenvalue associated with *f_k_*.

### 2.2. HawkFish Optimization Algorithm

The HawkFish Optimization Algorithm (HFOA) is a meta-heuristic algorithm proposed by Ali Alkharsan that simulates the sex-changing behavior of hawkfish and incorporates a dual fitness function [32]. During the parameter optimization process, this algorithm demonstrates exceptional robustness, efficiently managing volatility and noise interference in fault data. Simultaneously, HFOA ensures rapid convergence characteristics, thereby stably approaching the optimal solution within a relatively short time. The unique sex-changing process of hawkfish is as follows.

Initialize the male-to-female ratio of Eagle Fish:(3)p(t)+q(t)=1

In the equation p(t) represents the proportion of females at time t; q(t) represents the proportion of males at time t; at the initial moment, it is assumed that all eagle fish are female, p(t) = 1, q(t) = 0.

The rate of change in the sex ratio is as follows:(4)$$\begin{array}{l} \frac{dp}{dt}=-a(1-d(t))p\\ \frac{dq}{dt}=a(1-d(t))p \end{array}$$

In the equation d(t) represents food availability, where d(t) > 0 indicates sufficient food supply, and d(t) = 0 indicates scarcity; a represents the rate of gender change.

As indicated in Equation (4), the rate of change in the sex ratio is proportional to the current female ratio, food availability, and the rate of sex change. When food is abundant, the eagle fish population remains in equilibrium, with the sex ratio remaining constant over time. Conversely, when food is scarce, the population of eagle fish gradually shifts from being female-dominated to male-dominated. The process of sex change is illustrated in Figure 2, where *f*_1_ and *f*_2_ represent two distinct fitness functions.

#### Testing of the HFOA Algorithm

To validate the performance of the HFOA algorithm regarding global search capability, computational efficiency, and adaptability to dynamic environments, this study employs the benchmark function set as defined in Equation (5) and conducts comparative tests against the widely researched PSO and Grey Wolf Optimizer (GWO) algorithms. *F*_1_(*x*) is a complex convex function suitable for testing the global search capability of algorithms in high-dimensional spaces; *F*_2_(*x*) is a simple convex quadratic function used to evaluate the local search capability and convergence accuracy of algorithms; *F*_3_(*x*) is a non-smooth, non-convex function appropriate for testing algorithms’ ability to handle non-smooth characteristics and global search capabilities; *F*_4_(*x*) is a multi-modal non-convex function used to assess algorithms’ global and local search capabilities in complex non-convex optimization problems. These four test functions encompass optimization scenarios ranging from simple to complex, allowing for a comprehensive evaluation of optimization algorithms’ performance across different problem types, including unconstrained global optimization, convex optimization, non-smooth non-convex optimization, and multi-modal global optimization problems.

The results presented in Figure 3 (the number of iterations was uniformly set to 100) indicate that the proposed method surpasses the comparative algorithms in both convergence speed and solution accuracy. The HFOA demonstrates a faster convergence rate, allowing it to reach the optimal solution within a shorter time frame. Moreover, the solutions obtained through this method are closer to the global optimum than those produced by the other algorithms.(5)$$\begin{array}{c} F_{1}(x)=\sum_{i=1}^{n}{\left(\sum_{j=1}^{i}x_{j}\right)}^{2}\\ F_{2}(x)=\sum_{i=1}^{n}{\left(\left[x_{i}+0.5\right]\right)}^{2}\\ F_{3}(x)=\sum_{i=1}^{n}|x_{i}|+\underset{x\to \infty }{\lim}\prod_{i=1}^{n}|x_{i}|\\ F_{4}(x)=-20\exp\left(-\left(\frac{1}{5}\right)\sqrt{\frac{1}{n\sum_{i=1}^{n}x_{i}^{2}}}\right) \end{array}$$

### 2.3. MSCSA-Swin Transformer

#### 2.3.1. Swin Transformer

The Swin Transformer [33] is a hierarchical vision Transformer model specifically designed for vision tasks. Its primary feature is the implementation of the Shifted Window Multi-Head Self-Attention (SW-MSA) mechanism, which enhances feature representation capabilities while simultaneously reducing computational complexity. The network architecture consists of multiple stages that efficiently perform image feature learning through Hierarchical Feature Extraction (HFE) and Window Multi-Head Self-Attention Computation (W-MSA). Figure 4 illustrates the network architecture of the Swin Transformer model (the input variable is an H × W × 3 Patch Token; the output variable is a feature map of 32 H × 32 W × 768).

#### 2.3.2. Multi-Stage Cross-Scale Attention Mechanism

The Swin Transformer has demonstrated excellent performance in visual tasks; however, it has certain limitations. The local window restricts the modeling of global dependencies and lacks sufficient capability for capturing fine-grained features. Additionally, the hierarchical structure does not significantly enhance cross-stage feature fusion, and the multi-scale modeling lacks dynamic adaptability. This paper introduces a multi-stage cross-scale attention mechanism (multi-stage cross-scale attention, MSCSA) [34] to address the shortcomings of the Swin Transformer.

The MSCSA enhances feature modeling capabilities through multi-stage feature fusion, cross-scale attention computation, and the reinforcement of parallel convolutional paths. Initially, it collects feature maps from various stages, aligns them via pooling, and concatenates them to explicitly fuse multi-layer information. Subsequently, it generates key-value pairs at different resolutions through multi-scale projection, facilitating cross-scale interaction. Finally, parallel convolutional paths are introduced to strengthen local representations and address the limitations of self-attention in modeling short-range dependencies. Figure 5 illustrates the structure of MSCSA (the input variable is a multi-scale feature map with dimensions h × w × c; the output variable is a feature map of size 32 H × 32 W × 768).

(1) Multi-Scale Key-Value Generation

The input feature map X∈ℝh×w×c generates multi-scale features through different downsampling operations:(6)X0=XX1=DWConv1(X)X2=DWConv2(X)

In the equation, X∈ℝh×w×c represents the input feature image, with dimensions h×w×c. The operations DWConv1 and DWConv2 denote depth wise separable convolutions with downsampling factors of 2× and 3×, respectively. Each instance of Xi undergoes a linear projection to derive multi-scale key-value pairs:(7)Ki=XiWik, Vi=XiWiv

In the equation, $W_{i}^{k}$ and $W_{i}^{v}$ are the linear projection weight matrices, where *i* = 0, 1, 2 corresponds to different scales.

(2) Cross-Scale Attention Computation

Compute attention between query $Q=XW^{q}$ and the concatenated key $K=Concat(K_{0},K_{1},K_{2})$ and value $V=Concat(V_{0},V_{1},V_{2})$:(8)$$Attn(Q,K,V)=Softmax\left(\frac{QK^{T}}{\sqrt{d}}\right)V$$

In the equation, *d* represents the dimension of the attention head.

(3) Parallel Convolutional Pathways

To enhance local features, the original resolution values of the image are injected into the attention output through the convolutional path:(9)$$ModifiedAttn=Attn(Q,K,V)+DWConv(Hardswish(V_{0}))$$

In the equation: DWConv represents a 3 × 3 depth wise separable convolution; Hardswish denotes a nonlinear activation function.

To address the limitations of the standard Swin Transformer, this paper introduces an MSCSA module following each Transformer layer to enhance global modeling and fine-grained representation capabilities. The MSCSA conducts convolutional projection on both the original and downsampled feature maps, thereby integrating multi-scale information and facilitating global interaction through attention operations. Furthermore, a 3 × 3 depth wise separable convolution path is incorporated to bolster local perception, while an Intra-FFN is designed to reduce computational costs, achieving effective cross-stage and cross-scale feature collaboration. This module overcomes the bottleneck of local perception, fuses semantic information from both shallow and deep layers, and significantly enhances the model’s performance in classification tasks under limited resources.

### 2.4. ECA-Res2net

#### 2.4.1. Res2Net

Currently, many backbone architectures enhance their performance at the layer-wise level to improve the network’s ability to extract multi-scale features. Res2Net [35] refines this process at a more granular level by enhancing the internal structure of the residual block, thereby increasing the module’s capability to extract multi-scale features. The Bottleneck block in ResNet is illustrated in Figure 6a. Based on this structure, the architecture depicted in Figure 6b is constructed and referred to as the Res2Net module.

Res2Net introduces a fine-grained multi-scale feature processing approach, which fundamentally subdivides the traditional 3 × 3 convolution operation within residual blocks into multiple small-scale convolution groups by implementing a hierarchical residual connection mechanism. Unlike conventional methods that achieve multi-scale expression across network layers, Res2Net opts to introduce multi-scale paths within a single residual unit, significantly enhancing the CNN’s capability to respond to features of varying scales.

#### 2.4.2. Efficient Channel Attention

Although Res2Net enhances multi-scale feature representation, it has notable limitations. Its hierarchical residual structure lacks inter-channel interaction, which may suppress or inadequately fuse key features. As the scale parameter increases, the number of convolutional branches grows, leading to an increase in serial 3 × 3 operations that elevate computational latency, significantly impacting performance in resource-constrained environments. Furthermore, it employs a static residual weight allocation mechanism, which hinders the dynamic adjustment of each channel’s importance, resulting in diminished robustness when dealing with noise and complex backgrounds.

To address the aforementioned issues of Res2Net, the ECA attention mechanism [36] is introduced. ECA achieves cross-channel interaction through local one-dimensional convolution, thereby avoiding channel dimensionality reduction and significantly enhancing performance. By embedding ECA into Res2Net, an ECA module is integrated after each residual block to dynamically calibrate the multi-scale feature channels, thus improving the model’s focus on and utilization of key channels. This approach preserves the multi-scale feature extraction advantages of Res2Net while improving channel fusion efficiency, reducing complexity, and accelerating convergence speed. The structure diagram of the ECA attention mechanism is shown in Figure 7.

### 2.5. Introduction to HFMD and the Dual-Branch Parallel Network Model

This paper proposes a rolling bearing fault diagnosis method based on HFMD and a dual-branch parallel network. First, the HFMD algorithm is employed to decompose the original sound signal, alleviating the information aliasing problem caused by the sensitivity of modal parameters and structural disturbances. Subsequently, the optimal modes are selected and input into two feature extraction pathways: the time-domain channel integrates the Res2Net multi-scale residual structure with the ECA attention module to enhance local feature extraction capability; the time-frequency channel generates time-frequency maps through CWT, combined with a multi-stage cross-scale attention mechanism and Swin Transformer to model global dependencies across frequency bands. The dual-branch network achieves cross-modal feature alignment through collaborative learning and structural preservation, while the fusion layer enables pixel-level or channel-level interactions, significantly improving fault recognition ability, robustness, and model generalization performance in complex noise environments.

## 3. HFMD and Dual-Branch Parallel Network Fault Diagnosis Model

### 3.1. Validation of HFMD

FMD lacks the function of parameter self-adaptation, and manual parameter adjustment is unstable and inefficient. Among them, if the number of modes *n* is too small, the filtering results will be coarse, and potential fault information will be lost; if it is too large, the results will be distorted, thereby increasing the computational burden. If the filter length *L* is too small, the signal will be under-decomposed; if it is too large, the signal will be over-decomposed, increasing the running time and reducing computational efficiency. If the number of cutting frequency bands *K* is too large, it will generate excessive redundant information and increase the computational burden; if it is too small, some important signal features will be ignored, affecting the decomposition effect. If the cycle period *m* is too small, it will affect the convergence of the algorithm; if it is too large, it will increase the computation time. Therefore, this paper uses HFOA to optimize the combination of four key parameters, improving the decomposition effect while enhancing the decomposition efficiency. The implementation steps for HFMD are as follows:

(1) To initialize the population for the HFOA [32], set the population size *N* to 100, the spatial dimension *S* to 4, and the number of subgroups *k* to 3. The global learning coefficient is designated as α = 0.6, while the local learning coefficient is set to β = 0.7. The subgroup learning coefficient is established at *w* = 0.5. The maximum number of iterations is capped at 50. Additionally, the number of modes *n* is constrained within the range of [3, 10], the filter length *L* varies between [50, 300], the number of cut bands *K* is limited to [5, 12], and the cycle period *m* is defined within [5, 15], ensuring that *K* is greater than or equal to *n* [17].

(2) Based on the dynamic adaptability and rapid convergence of HFOA gender transformation, A dual fitness function is employed to evaluate individuals and select the optimal parameter combination. In this paper, *f*1 represents the envelope entropy, and *f*2 represents the crest factor (CF). A smaller envelope entropy indicates a more concentrated distribution of the reconstructed signal envelope, suggesting stronger signal regularity. The magnitude of the *CF* is directly proportional to the periodic transient components in the signal.

(3) Update individual positions based on fitness:(10)$$x_{i,j}=x_{i,j}+s_{i,j}d_{i,j}$$

In the equation, $x_{i,j}$ represents the *j*-th dimensional position of the *i*-th individual; $s_{i,j}$ denotes the step size; $d_{i,j}$ is the direction vector.

(4) To calculate the distance matrix among individuals in the population, we employ a clustering algorithm based on Euclidean distance. This method facilitates the division of the population into *k* distinct subpopulations. Subsequently, we identify the individual with the highest fitness within each subpopulation.

(5) Within each subpopulation, individuals update their positions based on the best positions of their neighbors.(11)$$x_{i,j}=x_{i,j}+w(x_{j\mathrm{best},j}-x_{i,j})$$

(6) Update the step size and direction vector based on the positions of both the global best individual and the local best individual.(12)$$\begin{array}{l} s_{i,j}=s_{i,j}+\alpha (x_{\mathrm{global},j}-x_{i,j})r\\ d_{i,j}=d_{i,j}+\beta (x_{i,j}-x_{\mathrm{local},j})r \end{array}$$

In the equation: α represents the global learning coefficient, β denote the local learning coefficient, r be a random number uniformly distributed between 0 and 1, $x_{\mathrm{global},j}$ signify the *j*-th dimensional position of the globally best fish, and $x_{\mathrm{local},j}$ indicate the *j*-th dimensional position of the locally best fish.

(7) When the number of iterations is less than the total number of iterations, repeat steps 2 through 6 until the iteration count meets the specified requirement, and subsequently output the optimal parameter combination.

(8) The minimum Residual Energy Index (REI) serves as the criterion for selecting the optimal modal component following the decomposition of High-Frequency Modal Data (HFMD).

To verify the signal processing capability of HFMD, the following simulation signal is constructed: (13)$$\left\{\begin{array}{l} x(t)=s(t)+n(t)\\ s(t)=\sum_{i}Ah(t-iT-\tau_{i})\\ h(t)=e^{-Ct}\sin(2\pi f_{n}t) \end{array}\right.$$

In the equation: s(t) represents the simulated periodic impact pulse signal; *A* is the amplitude of the simulated signal; h(t) is the oscillating signal with amplitude attenuation; n(t) is the added Gaussian white noise; $\sum_{i}$ indicates that h(t) is repeated *i* times until the sampling point requirement is met; *T* is the repetition period *T* = 0.0071, taken as *f* = 140 Hz, which is the fault characteristic frequency; *C* is the attenuation coefficient, let $f_{n}$ simulate the resonance frequency, set $f_{n}$ = 3000 Hz; the sampling frequency is 16 kHz; the number of analysis points is 16,000 points.

The optimization process was visualized to analyze the results of different algorithms in optimizing the key parameters of the FMD. HFOA possesses a unique gender transformation process. Therefore, for the dual fitness functions in this paper, PSO and GWO are optimized twice, respectively, and the fitness curves of each algorithm are as shown in Figure 8. Analyzing the envelope entropy and crest factor reveals that HFOA demonstrates exceptional performance in optimization speed and accuracy, successfully identifying the global optimal solution by the 16th iteration, with corresponding envelope entropy and crest factor values of 0.679 and 3.805, respectively. The envelope entropy indicates that PSO and GWO encountered new solutions multiple times throughout the iteration process; however, their optimization efficiency and accuracy were relatively low. Similarly, the crest factor analysis shows that both PSO and GWO exhibit insufficient optimization accuracy and struggle to escape local optima.

To verify the noise reduction effect of HFMD, a −15 dB noise level was added to the fault simulation signal described in Equation (13), as illustrated in Figure 9. Several noise reduction methods were compared, including PSO-VMD, fixed parameter FMD, fixed parameter VMD, and CEEMD. The time-domain diagrams and envelope spectra of the optimal components from different noise reduction methods are presented in Figure 10. The time-domain diagram of the optimal component of HFMD indicates that the majority of the noise has been effectively filtered out, and its envelope spectrum distinctly displays the 6th harmonic, which demonstrates a significant improvement over the other comparison methods. In contrast, the envelope spectrum of PSO-VMD reveals the 5th harmonic, with its decomposition results being superior to those of the other three comparison methods.

### 3.2. Dual-Branch Parallel Network Model

The model presented in this paper employs a dual-branch parallel input approach. The time-frequency graph branch integrates the MSCSA mechanism with the Swin Transformer, effectively addressing the challenge of inadequate global time-frequency feature capture in the original Swin Transformer when processing non-stationary noise. Meanwhile, the time-domain graph branch incorporates the ECA mechanism into the Res2Net multi-scale residual structure, thereby overcoming the limitations of traditional single-scale convolution in separating sparse features within acoustic signals. The structure of the dual-branch parallel network model is illustrated in Figure 11 (the input variables are the CWT time-frequency diagram and time-domain diagram processed by HFMD; the output variable is a fused feature of 32 H × 32 W × 1024), while the network hyperparameters are presented in Table 1, Table 2 and Table 3.

Explanation on the Basis of Key Module and Layer Configuration Settings:

(1) Res2Net + ECA Module (Temporal Branch): Res2Net achieves feature extraction for impact signals of varying widths through multi-scale subdivision within residual blocks; The ECA module is embedded after each Res2Net block to dynamically weight channels in a lightweight manner, enhancing responses of critical impact channels and improving noise immunity.

(2) MSCSA-Swin Transformer Module (time-frequency branch): While the original Swin Transformer possesses window-level attention capability, it struggles to process cross-scale frequency band information, particularly exhibiting insufficient perception under non-stationary noise conditions. We introduce the MSCSA (multi-stage cross-scale attention) mechanism to achieve multi-layer feature aggregation from Stage1 to Stage4, while incorporating parallel convolutional pathways to enhance the analysis capability for short-duration transient features.

(3) Parameter settings including layer counts and window sizes: the Swin Transformer adopts a standard hierarchical configuration of 2-2-6-2 across its stages, with a window size set to 7 and 4 × 4 patch partitioning. This architecture, widely validated in visual Transformers, is well-suited for spectrogram analysis. The Res2Net branch employs progressively increasing channel numbers and branch quantities (64–512), coupled with ECA to achieve dynamic response regulation across different scales, effectively balancing performance with computational efficiency.

Table 4 illustrates the data fusion process of the proposed method, clearly demonstrating the feature extraction procedure, feature-level fusion through channel concatenation, and the final classification processing.

### 3.3. Research Process of Bearing Fault Diagnosis Based on HFMD and Dual-Branch Parallel Network Under Acoustic Signals

To achieve a precise diagnosis of rolling bearing faults driven by acoustic signals, this paper proposes a diagnostic model based on HFMD and a dual-branch parallel network. The overall framework of this model comprises four modules, as illustrated in Figure 12. (The input variables are the CWT time-frequency diagrams and time-domain diagrams of different signals processed by HFMD; the output variable is the fault type category label).

(a) Construction of Multi-Physical Domain Rolling Bearing Systems for Physics-Data Fusion Modeling

Based on dynamic theory, a physical fault simulation model for bearings was constructed, integrating structural vibration with acoustic signal characteristics to enable controllable simulation of signal responses. Consequently, a bearing test platform was established to systematically collect actual sound signals during operation. By comparing the consistency between the physical simulation model and the actual system response, robust physical prior support was provided for the subsequent development of data-driven models.

(b) Cross-modal Feature Preservation Strategy for HFMD Data Decoupling and Redundant Modality Suppression

To address the modal drift and structural perturbation differences between the measured sound signals and simulation data, we propose a feature modal extraction model based on HFMD. By introducing the Eagle Bird Optimization Algorithm, we achieve structural consistency enhancement and fault-sensitive modal separation of data from different sources, thereby improving the identifiability and robustness of the feature space. Additionally, we perform preliminary noise reduction and preprocessing on the sound signals to extract the components that contain the most fault information, which serve as the data input for the dual-channel model verification in Module 3.

(c) Construction of Hierarchical Representation and Physical Perception Guidance Mechanism for Dual-Channel Collaborative Sensing Networks

This module establishes a fixed dual-channel input paradigm based on the optimal modal components of HFMD. The denoised time-domain diagram of HFMD is fed into the time-domain branch, which integrates Res2Net multi-scale residuals and ECA channel attention. Meanwhile, the time-frequency diagram transformed by CWT is input into the time-frequency branch, incorporating MSCSA and the Swin Transformer. The rationale for fixing the channel mapping is that, while the standard Swin Transformer can efficiently perform self-attention within local windows, it lacks the ability to establish cross-window multi-scale band associations. This limitation makes it challenging to capture long-range dependencies and transient spectral mutations under non-stationary noise backgrounds. Although the standard Res2Net possesses multi-scale temporal decomposition capabilities, it cannot adaptively adjust sub-scale channel weights, resulting in insufficient resolution of weak impact pulses. MSCSA achieves multi-layer aggregation of global band information by establishing dynamic channel space interactions across different attention heads and levels. ECA, with its lightweight channel re-calibration mechanism, dynamically enhances key scale components while suppressing noisy sub-channels. The dual-branch architecture facilitates cross-modal alignment under the constraints of collaborative feature learning and structural preservation, thereby maximizing the synergy between global time-frequency modeling and local temporal resolution. This significantly enhances the sensitivity of fault signal perception and the robustness of diagnosis.

(d) Model Generalization Capability Evaluation and Decision Robustness Analysis Based on Multi-Domain Consistency Verification

This study constructs a multi-indicator performance evaluation system for bearing fault classification and systematically assesses the proposed dual-channel model across dimensions such as accuracy, generalization capability, and stability. By conducting cross-testing and ablation experiments with both open-source datasets and measured data, we verify the model’s robustness and diagnostic reliability under multi-source heterogeneous data, thereby providing theoretical and experimental support for its engineering promotion.

The motivation for combining HFMD with the dual-branch parallel neural network model stems from a comprehensive approach to address the issues of sparse features, information aliasing, and insufficient cross-modal feature representation capability in acoustic signals of rolling bearings.

(1) Introduction of HFMD

The acoustic signal characteristics are complex: Rolling bearings generate non-stationary and nonlinear acoustic signals during operation, which are easily affected by environmental noise under actual working conditions. Although traditional signal decomposition methods can decompose features to a certain extent, they still face issues such as mode mixing and strong parameter dependence.

The performance of FMD is limited by parameter selection: Although FMD has advantages in noise suppression and periodic impact capture, its performance is highly dependent on the choice of parameters. To achieve more stable signal decomposition, HFOA is introduced to optimize the key parameters of FMD, resulting in the construction of HFMD, which possesses stronger adaptability and stability, effectively extracting the optimal modal components that contain fault information.

The optimized HFMD can effectively improve the signal-to-noise ratio and feature separability. The optimal modes selected by REI not only retain critical fault information but also reduce noise interference, providing clearer and more distinguishable input data for subsequent deep learning models.

(2) Introduction of the Dual-Branch Parallel Model

Limited feature extraction capability from a single perspective: Most scholars’ fault diagnosis models predominantly feature a single-branch structure, which typically processes only one type of feature representation. This makes it challenging to simultaneously address local temporal patterns and global frequency band relationships.

The necessity of multi-scale and multi-modal feature extraction: This study designs a dual-branch structure operating in both the time domain and the time-frequency domain. The time-domain branch is based on a multi-scale residual structure of Res2Net, combined with the ECA attention mechanism, which enhances the ability to capture shock pulses and local patterns. The time-frequency domain branch employs CWT to convert signals into time-frequency images, integrating the Swin Transformer model of MSCSA to model the dependencies between global frequency bands, thereby improving the representation capability for complex signals.

(3) Overall Integration and Advantages

The integration of HFMD with a dual-branch model embodies the concepts of high-quality input and collaborative modeling. HFMD is responsible for cleaning noise from the signal level and extracting optimal modal components; the dual-branch network captures multidimensional information from the feature level, enabling precise identification of bearing faults.

In summary, HFMD provides clearer and more diagnostically valuable inputs for deep networks, while the dual-channel structure ensures sufficient expression and fusion of multi-scale features. The combination of these two approaches has achieved significant performance improvements in fault diagnosis.

## 4. Open-Source Data Verification

### 4.1. QU-DMBF Test Bearing and Test Bench

The bearing dataset from Qatar University (QU-DMBF, open-source dataset) [37] is presented in Figure 13. The test bench is powered by a DC motor with specifications of 0.37 kW, 90 VDC, and 5 A. Data acquisition is conducted using the NI-9234, which collects fault data at a sampling frequency of 4.096 kHz.

The QU-DMBF bearing dataset employs the NSK-6208 model of rolling bearings, from which acoustic signals are collected under various operating conditions. Bearing faults are introduced using electrical discharge machining. In this experiment, a load of 0.18 kN and a rotational speed of 240 RPM were selected, with single-point faults introduced with diameters of 0.35 mm and 0.4 mm for the inner and outer raceway faults, respectively. The NSK-6208 has a pitch circle diameter of 60 mm, a rolling element diameter of 7.38 mm, a rolling element count of 9, and a contact angle of 0°. The calculated inner raceway fault frequency is 20.38 Hz, while the outer raceway fault frequency is 15.62 Hz.

### 4.2. Verification of HFMD Denoising Effect

A comparison between HFMD and PSO-VMD is presented, with the fitness function curves of different methods shown in Figure 14. HFMD quickly finds the global optimal solution by the 18th iteration, achieving corresponding envelope entropy and crest factor fitness values of 0.6883 and 3.6912, respectively. In terms of optimization speed and accuracy, HFMD outperforms the well-performing signal decomposition method, PSO-VMD, which further validates the advantages of HFOA in parameter optimization.

Both HFMD and PSO-VMD were employed to denoise the QU-DMBF data, and the envelope spectra of the optimal components are presented in Figure 15. From Figure 15a,c,e,g, it is evident that the envelope spectra of the acoustic signals processed by HFMD clearly reveal the inner raceway fault frequency and its multiple harmonic components, with the harmonic amplitudes being notably prominent. This sufficiently indicates that the bearing has developed a fault. In contrast, the analysis of Figure 15b,d,f,h shows that in the envelope spectra of PSO-VMD, only the fundamental frequency is observable, while its second and third harmonic components are not distinct. This indicates limited denoising effectiveness, with issues of spectral line splitting and mode aliasing still persisting.

### 4.3. Introduction to the QU-DMBF Dataset

Table 5 presents the fault classification of the QU-DMBF dataset, which includes five groups of acoustic data corresponding to samples of normal bearings, as well as inner and outer raceway faults with fault diameters of 0.3 mm and 1.0 mm. The QU-DMBF dataset comprises 300 data samples for each type of bearing fault, amounting to a total of 1500 samples. These data samples are partitioned into training, validation, and test sets in a 6:2:2 ratio.

### 4.4. MSCSA-Swin Transformer and ECA-RES2net Verification and Analysis

In the QU-DMBF dataset, data samples are acquired using a step sampling method, resulting in a total of 2,252,740 acoustic data points, with every 2048 data points constituting a sample. A sliding window approach is employed, where adjacent samples overlap by 548 data points, and the step size of the sliding window is set to 1500 data points. The CWT [38] is utilized to process the optimal component following HFMD denoising. Through experimentation, the Daubechies5 wavelet is selected as the basis function, with the total number of scales set to 1024. The scale sequence is constructed based on the wavelet’s center frequency, thereby generating time-frequency diagrams that serve as the input for the MSCSA-Swin Transformer. The time-domain diagrams of the optimal components extracted from HFMD serve as inputs for ECA-RES2net. The time-domain and time-frequency diagrams of various signals are illustrated in Figure 16.

To verify the diagnostic efficacy of the parallel branch network model proposed in this paper, as well as the effectiveness of MSCSA on the Swin Transformer and ECA on RES2net, ablation experiments were conducted. The parallel branch network model for the Swin Transformer and RES2net is denoted as S-R, while MSCSA applied to the Swin Transformer and RES2net is referred to as MS-R. The combination of the Swin Transformer and ECA-RES2net is labeled as S-ER, and the model proposed in this study is designated as MS-ER. Figure 17 presents the experimental results of each model on the QU-DMBF dataset. In the loss curves depicted in Figure 17a,c, S-R experiences a rapid decline after the 29th and 32nd rounds, converging at 0.4236 and 0.5143, respectively. In contrast, MS-R and S-ER exhibit a swift descent and tend to converge within the first 20 iterations. The model proposed in this study, MS-ER, converges at 0.0384 and 0.0421 after the 11th iteration, with its loss significantly lower than that of the other models. In the accuracy convergence curves of the training and validation sets shown in Figure 17b,d, the S-R model displays a slow upward trend during the first 23 iterations, with its accuracy converging at 93.56% and 94.33%, respectively. This is significantly lower than the improved models MS-R and S-ER. Moreover, the accuracy of the MS-ER training and validation sets rises rapidly, converging to 100% by the 12th iteration.

Table 6 presents the diagnostic results of different models on the test set. The proposed MS-ER model achieves a 100% fault identification accuracy on the QU-DMBF open dataset, significantly outperforming the other four methods. However, to further validate its computational resource efficiency, this paper conducts a supplementary analysis of the parameter count and FLOPs of each model. The results show that the MS-ER has a parameter count of 3.95 M and FLOPs of 2.80 G, which is slightly higher than the simplified structures S-R (1.45 M, 0.98 G) and MS-R (3.12 M, 2.04 G). Nevertheless, given its significantly enhanced identification capability, the computational complexity remains within an acceptable range. Furthermore, compared to the S-ER model (2.98 M, 1.89 G), the MS-ER introduces a dual-branch design and attention mechanism, achieving an accuracy improvement of nearly 3.5% with only an increase of 0.97 M in parameters and less than 1 G in FLOPs. This further highlights its structural efficiency. Thus, it can be seen that the MS-ER model achieves a high degree of balance between computational performance and diagnostic performance, demonstrating strong deployment adaptability and engineering controllability.

### 4.5. CWRU Data Verification and Analysis

In addition to the acoustic signal experiments, we further validated the proposed model on the commonly used Case Western Reserve University (CWRU) rolling bearing fault dataset for a more intuitive comparison with existing research results both domestically and internationally. This dataset is widely applied in the field of vibration signal analysis and possesses good standardization and generality. The selected drive-end bearing for the experiment is the SKF 6205-2RS JEM deep groove ball bearing, operating at a speed of 1797 RPM and a sampling frequency of 12 kHz. We chose a single-point fault condition formed by electrical discharge machining, with a fault diameter of 0.1778 mm. From the CWRU dataset, a total of 1000 samples were used, with 250 samples each for four states of bearings: healthy, inner race fault, outer race fault, and rolling element fault. The data samples were divided into training, validation, and test sets in a ratio of 6:2:2. The data division of the CWRU dataset is illustrated in Table 7.

In order to comprehensively evaluate the diagnostic performance of the proposed method under different fault types, a comparative analysis was conducted with six models: (1) STFT-2DCNN [39]. The original signal is converted into a 2D time-frequency map using Short-Time Fourier Transform (STFT), which is then input into a 2DCNN for training. (2) CWT-2DCNN [40]. The original signal is transformed into a CWT time-frequency map, which is subsequently input into a 2DCNN for training. (3) CNN-GRU. The optimal components from the Hilbert-Huang Modal Decomposition (HFMD) are used as input in both the time domain and the CWT time-frequency map; CNN processes the CWT time-frequency map, while GRU handles the time-domain graph. (4) Dual channel 2DCNN. The data source is the same as in (2), where both branches analyze the CWT time-frequency map. (5) BiLSTM-ResNet [41]. The data source is the same as in (2). (6) Transformer-TELM [42]. The time-domain graph from the optimal components of HFMD is used as input. For the sake of analysis and discussion, the model presented in this paper will be referred to as MS-ER.

To avoid the randomness of the experiments, each model was tested 5 times, and the average values were taken. The final experimental results of different models on the test set are presented in Table 8. The MS-ER model significantly outperforms other classical and deep architectures in terms of accuracy, achieving 98.8%. Compared to Transformer-TELM (4.90 M, 3.95 G) and BiLSTM-ResNet (4.20 M, 3.60 G), the MS-ER model reduces the number of parameters and computational load by approximately 19% and 29%, respectively, while achieving about a 4% increase in accuracy. This synergistic optimization of performance and efficiency results from the effective compression of redundant paths by the cross-scale attention mechanism in the MSCSA module, as well as the efficient modeling capability of key channels under lightweight conditions provided by the ECA module.

The fault diagnosis method based on HFMD and a dual-branch parallel structure proposed in this paper has achieved significant results in diagnostic effectiveness. To further reveal the driving factors behind the model’s decisions, the XGBoost feature importance ranking has been introduced, as shown in Figure 18. By incorporating the XGBoost model for quantitative assessment of key features, the relative contributions of each feature to the diagnostic results can be clearly identified.

## 5. Acoustic Signal Test Verification and Analysis

To further validate the effectiveness of the HFMD method in acoustic signal denoising, as well as the exceptional performance of the MSCSA-Swin Transformer and ECA-RES2net in fault diagnosis, this paper conducts experimental verification and evaluation by collecting real bearing acoustic signal data (internally collected dataset).

### 5.1. Test Bench and Bearings

The test bench utilized in the experiment is the L38B model from Schaeffler Trading (Shanghai, China) Shanghai Co., Ltd.—Shanghai R&D Center. Figure 19a displays the test rig, while Figure 19b presents its structural diagram, featuring the test bearings symmetrically positioned on either side of the support bearings. During the experiment, the spindle speed is regulated via a belt drive, and the load is applied to the test bearings through disc springs. The fixture can be interchanged to accommodate various bearing models for testing. Figure 19c depicts the installation position of the acoustic sensor, which comprises an AWA14423 microphone and an AWA14604 preamplifier. Figure 19d illustrates the components of the acoustic sensor.

The test bearing model is FAG NU218-E-XL-TVP2, featuring an inner raceway diameter of 90 mm, an outer raceway diameter of 160 mm, a rolling element diameter of 13 mm, a rolling element length of 13 mm, and a total of 18 rolling elements. The data acquisition card utilized is the NI 9234, which is employed to collect bearing acoustic data at a sampling frequency of 51.2 kHz, with a total of 51,200 sampling points. The test speed is maintained at a constant rate of 300 r/min, while a radial load of 10 kN is applied. The fault frequency is calculated based on the dimensional parameters, as detailed in Table 9.

### 5.2. Verification and Analysis of HFMD Under Test Acoustic Signals

Figure 20 presents microscopic images depicting the healthy state, inner raceway fault, outer raceway fault, and rolling element fault, with specific fault locations highlighted within red boxes. Acoustic signals corresponding to these four fault types were collected for comparative analysis. Figure 21 illustrates the acoustic signals under various fault conditions. Each fault type exhibits distinct waveforms, varying signal amplitudes, and unique signal characteristics.

Different fault types of data were decomposed using HFMD for signal decomposition, with the minimum Residual Energy Index (REI) serving as the selection criterion for the optimal modal component. The REI values for each modal component corresponding to different bearing fault types are presented in Table 10. The optimal components identified for inner raceway fault, outer raceway fault, and rolling element fault are IMF4, IMF4, and IMF5, respectively. The envelope spectra of these optimal components for each fault are illustrated in Figure 22. Analysis reveals that the fault signals exhibit significant peaks at multiples of the fault frequency, clearly indicating the presence of corresponding faults in the bearing, which aligns with the actual situation. The Hilbert-Huang Transform demonstrates excellent feature extraction and anti-interference capabilities for the acoustic data in this experiment, effectively extracting the fault characteristics of the original signal and enhancing the diagnostic capability of the subsequent dual-branch parallel network.

### 5.3. Validation and Analysis of HFMD-MSCSA-Swin Transformer and ECA-RES2net

#### 5.3.1. Parallel Network Model Data Processing

The experimental acoustic signals, after processing by HFMD, have their optimal components extracted for the CWT time-frequency and time-domain diagrams, which serve as inputs to the MSCSA-Swin Transformer and ECA-RES2net. Figure 23 illustrates the time-domain diagrams and CWT time-frequency diagrams for different faulty bearings.

#### 5.3.2. Introduction to the Test Dataset

To fully capture the fault information of the bearings, data samples were collected using a step sampling method. Each sample consists of 5120 data points, with an interval of 1280 data points between adjacent samples, resulting in interval samples. A total of 1000 samples were obtained, with 250 samples corresponding to each bearing condition. The data samples were then divided into training, validation, and test sets in a ratio of 6:2:2. The distribution of the dataset is presented in Table 11.

#### 5.3.3. Comparative Analysis of Different Diagnostic Models

To validate the superiority of the MSCSA-Swin Transformer and ECA-RES2net parallel network models, experiments and analyses were conducted using the comparative models described in Section 4.5, which include STFT-2DCNN, CWT-2DCNN, CNN-GRU, and dual-channel 2DCNN.

Figure 24 illustrates the iteration curves of the loss function and accuracy for STFT-2DCNN, CWT-2DCNN, CNN-GRU, dual-channel 2DCNN, and the proposed method after 100 iterations on both the training and validation sets. It is evident from Figure 24 that the training speed and convergence of the MSCSA-Swin Transformer and ECA-RES2net models presented in this study are superior to those of the other models. On both the training and validation sets, the loss function converges to 0.0016 and 0.0236 in fewer than 10 iterations, while the accuracy stabilizes early, reaching 99.67% and 99.56%, respectively, which is significantly better than the other comparative models. The CNN-GRU and Dual-channel 2DCNN, which utilize a dual-branch parallel network model akin to the one proposed in this paper, undergo HFMD noise reduction. They demonstrate commendable performance in terms of loss function and accuracy, stabilizing within 10–20 iterations; however, neither surpasses the model introduced in this study. The performance of CWT-2DCNN significantly exceeds that of STFT-2DCNN. Unlike STFT, which suffers from energy leakage and fragmented fault impact information, CWT dynamically tracks transient impacts with a clear energy focus and easily identifiable sideband stripes. The superiority of the model presented in this paper for diagnosing bearing acoustic signals has been thoroughly validated.

To ensure the scientific validity and rationality of the experiment, each model was tested 5 times with the average results taken. Table 12 presents the diagnostic results of different models in the test set. In terms of parameter scale, the STFT-2DCNN and CWT-2DCNN have parameter counts of 1.12 M and 1.34 M, respectively, with FLOPs of 0.66 G and 0.84 G. Despite their relatively low computational cost, their accuracy (74.2% and 82.2%) is significantly lower than that of other models, making it difficult to meet the demands for high-reliability diagnosis. The CNN-GRU and dual-channel 2DCNN introduce temporal modeling and dual-path designs, respectively, resulting in significant improvements in accuracy (93.5% and 94.2%). However, this comes at the cost of increasing parameters and FLOPs to 3.22 M/2.85 G and 3.75 M/2.92 G, respectively. In contrast, the MS-ER achieves an accuracy of 98.4% with only a 0.2 M increase in parameters and less than a 0.1 G increase in FLOPs, demonstrating superior diagnostic performance.

#### 5.3.4. Diagnostic Model Ablation Test

To further validate the effectiveness of the proposed method, ablation experiments were conducted, comparing models that include the single-branch model and the dual-branch parallel model. The models evaluated are (1) The original signal is directly used as the input for ECA-RES2net, denoted as O-E-R. (2) The CWT time-frequency diagram of the original signal is used as the input for MSCSA-Swin Transformer, denoted as O-M-S. (3) The time-domain diagram of the optimal component, after HFMD processing of the original signal, is used as the input for ECA-RES2net, denoted as H-E-R. (4) The CWT time-frequency diagram of the optimal component, after HFMD processing of the original signal, is used as the input for MSCSA-Swin Transformer, denoted as H-M-S. After processing by HFMD, the time-domain plot and CWT time-frequency plot of the optimal component are used as inputs to the standard Swin Transformer-RES2net dual-branch model, denoted as H-S-R (6). The model proposed in this paper is referred to as H-MS-ER. A total of five datasets were designed: Dataset A (original signal images), Dataset B (CWT time-frequency images of the original signal), Dataset C (time-domain images after HFMD processing), Dataset D (CWT time-frequency images after HFMD processing), and Dataset E (time-domain and CWT time-frequency images after HEMD processing). The simplified diagrams of each model structure are shown in Figure 25.

To avoid randomness in the experiment and ensure scientific validity, the experiment was conducted five times, and the average value was computed. The final results are presented in Figure 26. The outcomes of the five experiments demonstrate that the diagnostic performance of the H-MS-ER model significantly surpasses that of all other models, achieving an average accuracy rate of 98.6%. Compared to the accuracy rates of O-E-R, O-M-S, H-E-R, H-M-S, and H-S-R, it shows improvements of 4.6%, 10.1%, 13.8%, 15.7%, and 20.3%, respectively. Notably, the accuracy rate of the dual-branch model exceeds that of all single-branch models, indicating that dual-input can capture more feature information in the network model than single-input, thereby enhancing recognition and classification effectiveness. The average accuracy of H-MS-ER increased by 4.6% compared to H-S-R, suggesting that the Swin Transformer and RES2net models, supported by MSCSA and ECA, can achieve superior diagnostic performance. Additionally, the average accuracy of H-E-R and H-M-S improved by 6.5% and 5.6%, respectively, compared to O-E-R and O-M-S, demonstrating that HFMD effectively eliminates interference components from the original test signals and can efficiently extract fault characteristics from these signals. Through the aforementioned experimental analysis and comparison, it is convincingly demonstrated that the dual-branch parallel network model designed in this paper is both rational and effective.

#### 5.3.5. Validation and Analysis of Diagnostic Models Under Different Sample Sizes

In deep learning models, the effectiveness of fault diagnosis varies with different input image samples. Particularly, when the data samples are limited, the model may experience overfitting due to the excessive number of parameters in the network. Therefore, it is crucial for the network to maintain good diagnostic performance even with fewer samples.

This paper selects 80%, 60%, 40%, and 20% of the total sample size of 1000, specifically 800, 600, 400, and 200 samples, to validate the superiority of the dual-branch parallel network model. The experiment was conducted 10 times, and the average values were calculated, with the results presented in Figure 27. Leveraging the excellent signal processing capability of HFMD, along with the robust fault recognition advantages of HFMD-MSCSA-Swin Transformer and ECA-RES2net, the accuracy of the training set, validation set, and test set of the proposed model remains above 95% as the sample size decreases.

To demonstrate the stability of the model’s diagnostic effectiveness, Figure 28 illustrates the rate of change in test set accuracy relative to a sample size of 1000 across various sample sizes. Although accuracy decreases as the sample size reduces, the rate of change generally stabilizes and does not increase sharply. For a sample size of 200, the rate of change is only 2.34%, and compared to the sample size of 800, the test set’s rate of change has increased by merely 1.57%.

To provide a more detailed demonstration of the diagnostic results for HFMD-MSCSA-Swin Transformer and ECA-RES2net, the initial trial results of bearing states under varying sample sizes were visualized, with the confusion matrix presented in Figure 29. As the sample size decreases, the model effectively identifies and accurately classifies various types of faults, achieving accuracy rates of 97.50%, 96.88%, 96.67%, 96.25%, and 95.00% at sample sizes of 1000, 800, 600, 400, and 200, respectively.

## 6. Conclusions and Prospects

### 6.1. Conclusions

To address the issue of varying data quality across different source domains, which arises from sparse feature separation in rolling bearing sound signals and subsequently affects fault diagnosis accuracy, this study proposes a rolling bearing fault diagnosis method based on HFMD and a dual-branch parallel network. Firstly, HFMD is employed to extract both the time-domain graph and the CWT time-frequency graph of the optimal modal components. Subsequently, these graphs are fed into the dual-branch parallel network model, which consists of ECA-RES2net and MSCSA-Swin Transformer, to obtain comprehensive fault diagnosis results. Finally, the accuracy and superiority of this method are validated using the QU-DMBF and experimental acoustic datasets. The research findings indicate that:

(1) Construction of the HFMD signal preprocessing algorithm: The HFOA is employed to optimize the four key parameters of the FMD, namely the number of modes *n*, filter length *L*, number of frequency bands *K*, and cycle period *m*. This optimization facilitates the extraction of fault information from the original signal, which is beneficial for subsequent model fault identification.

(2) Development of the MSCSA-Swin Transformer and ECA-RES2net dual-branch parallel network: The two branches achieve cross-modal alignment under the constraints of collaborative feature learning and structural preservation, maximizing the synergistic effects of global time-frequency modeling and local time-domain resolution. This significantly enhances the sensitivity of fault signal perception and the robustness of diagnostics.

(3) Multi-index performance evaluation of the HFMD and dual-branch parallel network model: A systematic evaluation of the proposed dual-branch parallel network model is conducted against other advanced models in terms of accuracy, generalization ability, and stability, using both open-source and experimental datasets.

### 6.2. Prospects

In light of the challenges encountered during this research process and the directions for future work, the following outlook is presented.

Limitations of this study:

(1) Although the proposed HFMD and dual-branch parallel network can maintain a diagnostic accuracy of over 95% with a limited sample size, and the variation in accuracy is only 2.34%, demonstrating good stability, the experimental results indicate that there is still a certain decline in accuracy as the sample size decreases. This suggests that the model experiences some performance fluctuations in extremely low sample scenarios.

(2) This study is relatively dependent on the decomposition effect of HFMD and the quality of the CWT transformation during the signal preprocessing phase, which places high demands on the rationality of algorithm parameter settings. Improper parameter settings may adversely affect the final feature extraction results and diagnostic accuracy.

Future work:

(1) It is essential to explore the introduction of methods such as meta-learning or transfer learning to enhance the model’s generalization ability and diagnostic accuracy under extremely small sample data.

(2) The robustness of the model should be validated in more complex and variable working conditions to further enhance its adaptability and generalization capability to multi-source heterogeneous acoustic data.

## Figures and Tables

**Figure 1 sensors-25-05338-f001:**
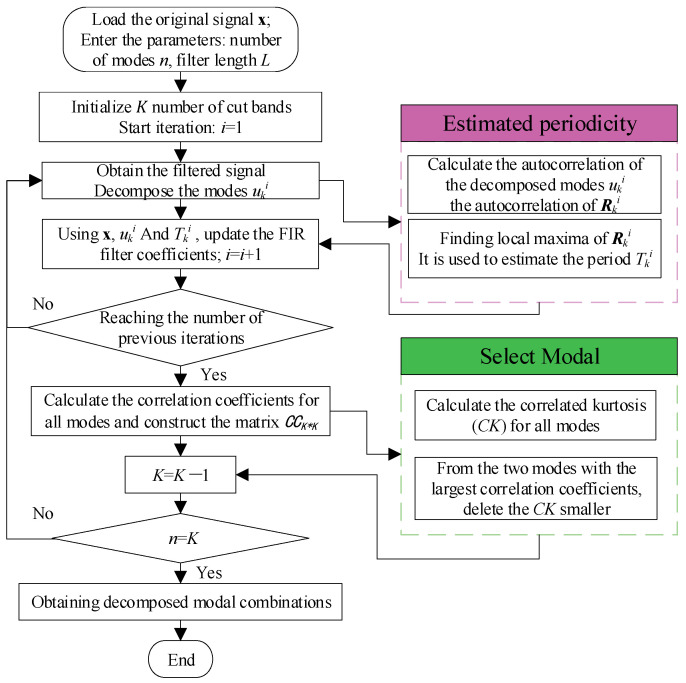
Flowchart of the FMD decomposition process.

**Figure 2 sensors-25-05338-f002:**
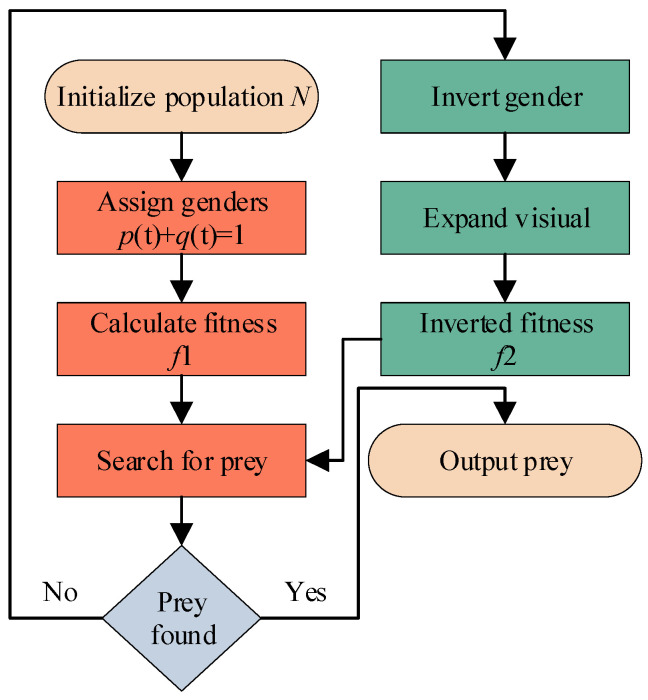
Gender transformation process of the Hawkfish.

**Figure 3 sensors-25-05338-f003:**
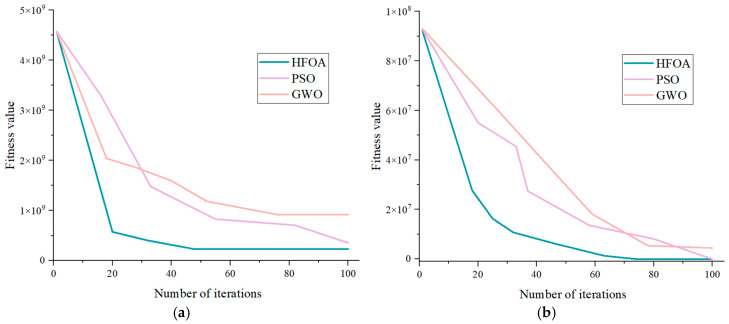
Convergence curves of the fitted values for the selected functions under different optimization methods. (**a**) Convergence curve of the F1 test function; (**b**) convergence curve of the F2 test function; (**c**) convergence curve of the F3 test function; (**d**) convergence curve of the F4 test function.

**Figure 4 sensors-25-05338-f004:**
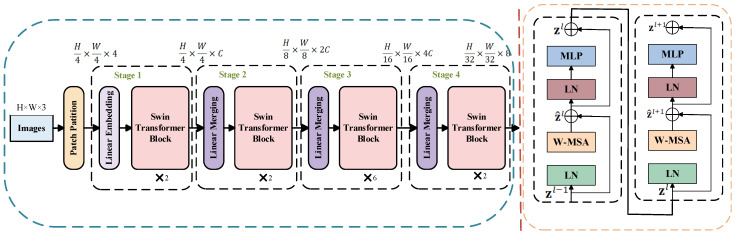
Model architecture diagram of Swin Transformer.

**Figure 5 sensors-25-05338-f005:**
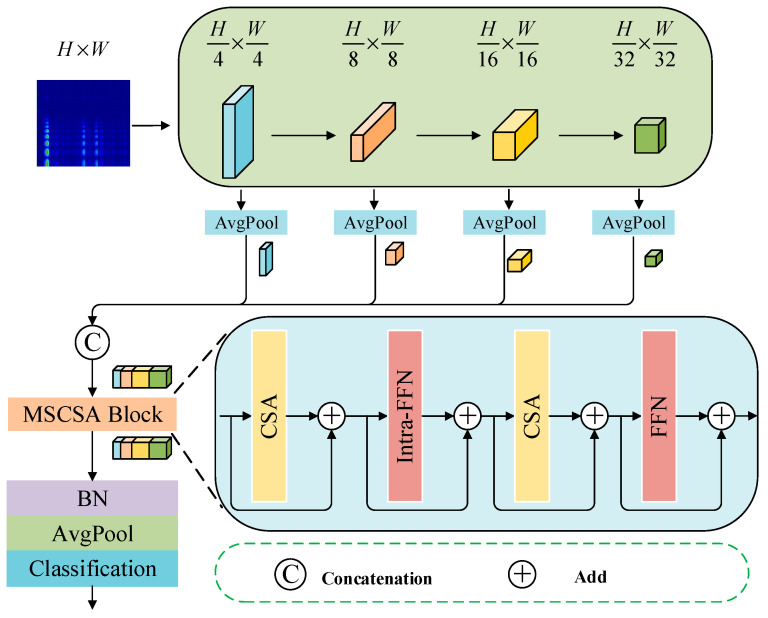
Structural diagram of MSCSA.

**Figure 6 sensors-25-05338-f006:**
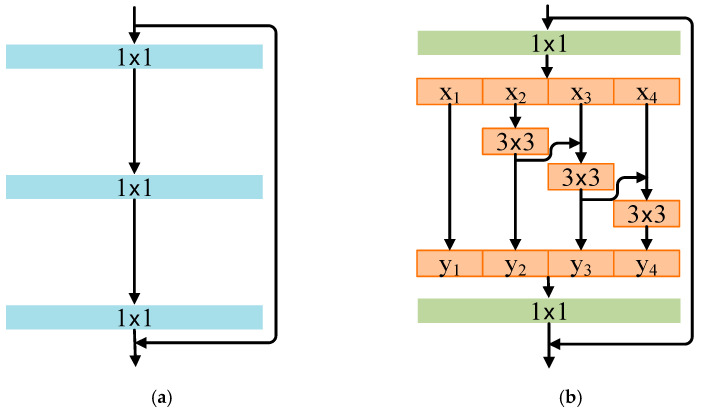
Construction of Res2Net. (**a**) Structural diagram of bottleneck; (**b**) structural diagram of Res2Net module.

**Figure 7 sensors-25-05338-f007:**
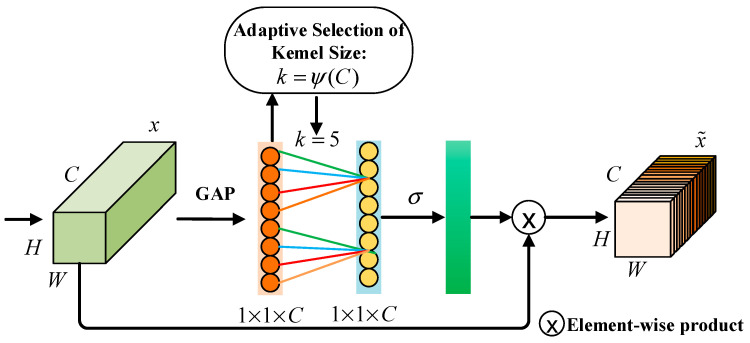
Structure diagram of the ECA attention mechanism.

**Figure 8 sensors-25-05338-f008:**
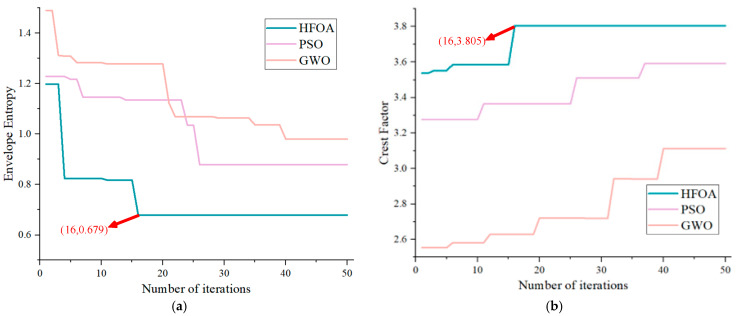
Fitness curves of FMD optimized by different methods. (**a**) Iteration curve of envelope entropy; (**b**) iteration curve of crest factor.

**Figure 9 sensors-25-05338-f009:**
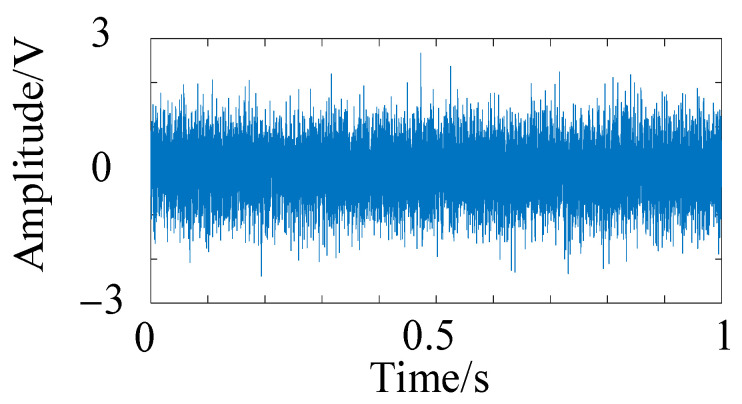
Time-domain diagram of the simulated signal.

**Figure 10 sensors-25-05338-f010:**
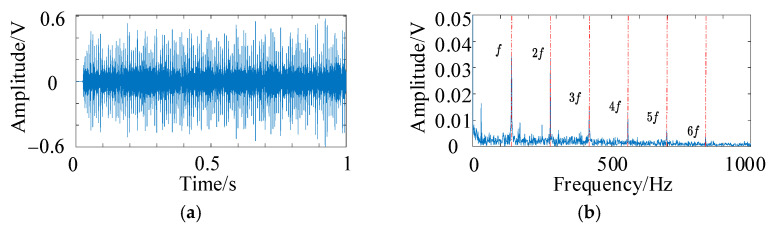
Time-domain diagrams and envelope spectra of the optimal components for different noise reduction methods. (**a**) The time-domain graph of HFMD; (**b**) the envelope spectrum of HFMD; (**c**) the time-domain graph of PSO-VMD; (**d**) the envelope spectrum of PSO-VMD; (**e**) the time-domain graph of fixed parameter FMD; (**f**) the envelope spectrum of fixed parameter FMD; (**g**) the time-domain graph of fixed parameter VMD; (**h**) the envelope spectrum of fixed parameter VMD; (**i**) the time-domain graph of CEEMD; (**j**) the envelope spectrum of CEEMD.

**Figure 11 sensors-25-05338-f011:**
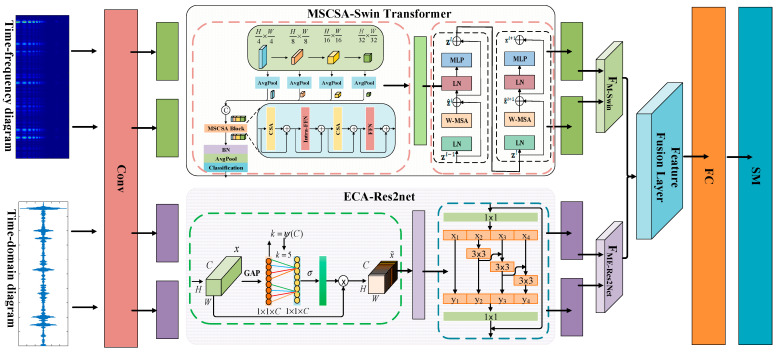
Diagram of the dual-branch parallel network structure.

**Figure 12 sensors-25-05338-f012:**
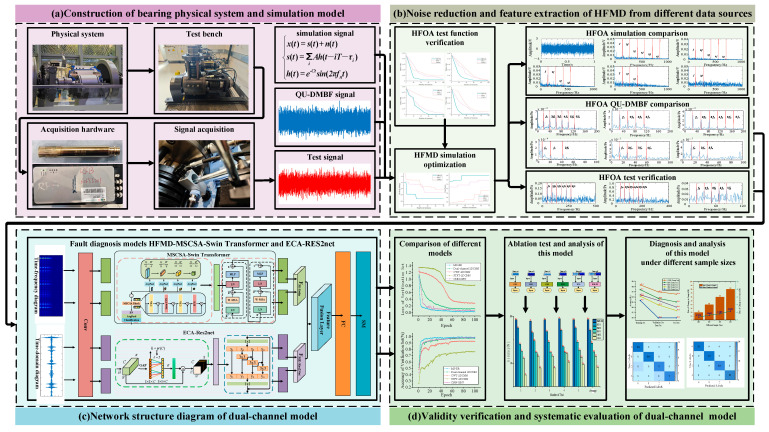
HFMD-Swin Transformer-Res2Net. (**a**) construction of bearing physical system and simulation model; (**b**) noise reduction and feature extraction of HFMD from different data sources; (**c**) network structure diagram of dual-channel model; (**d**) validity verification and systematic evaluation of dual-channel model.

**Figure 13 sensors-25-05338-f013:**
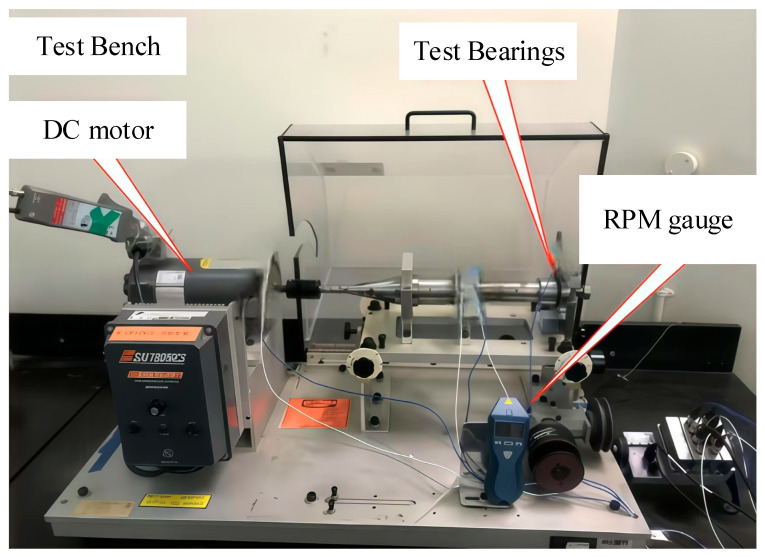
Bearing test bench of QU-DMBF.

**Figure 14 sensors-25-05338-f014:**
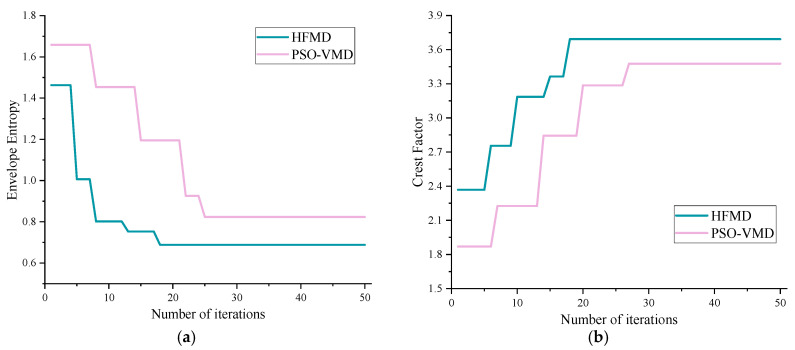
The iteration curve of HFMD and PSO-VMD. (**a**) Iterative curve of envelope entropy; (**b**) iterative curve of the crest factor.

**Figure 15 sensors-25-05338-f015:**
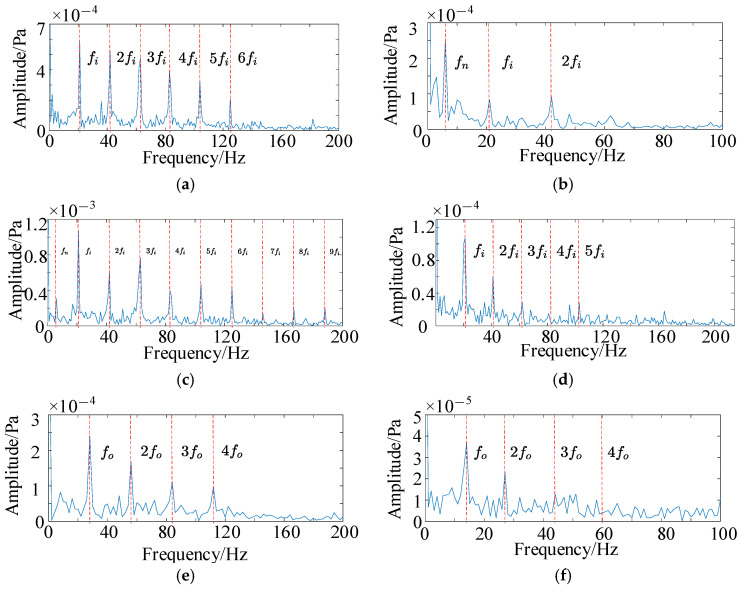
Envelope spectrum of the optimal ccomponents of HFMD and PSO-VMD. (**a**) Optimal component envelope spectrum of HFMD inner raceway fault (0.35 mm); (**b**) optimal component envelope spectrum of PSO-VMD inner raceway fault (0.35 mm); (**c**) optimal component envelope spectrum of HFMD inner raceway fault (0.40 mm); (**d**) optimal component envelope spectrum of PSO-VMD inner raceway fault (0.40 mm); (**e**) optimal component envelope spectrum of HFMD outer raceway fault (0.35 mm); (**f**) optimal component envelope spectrum of PSO-VMD outer raceway fault (0.35 mm); (**g**) optimal component envelope spectrum of HFMD outer raceway fault (0.40 mm); (**h**) optimal component envelope spectrum of PSO-VMD outer raceway fault (0.40 mm).

**Figure 16 sensors-25-05338-f016:**
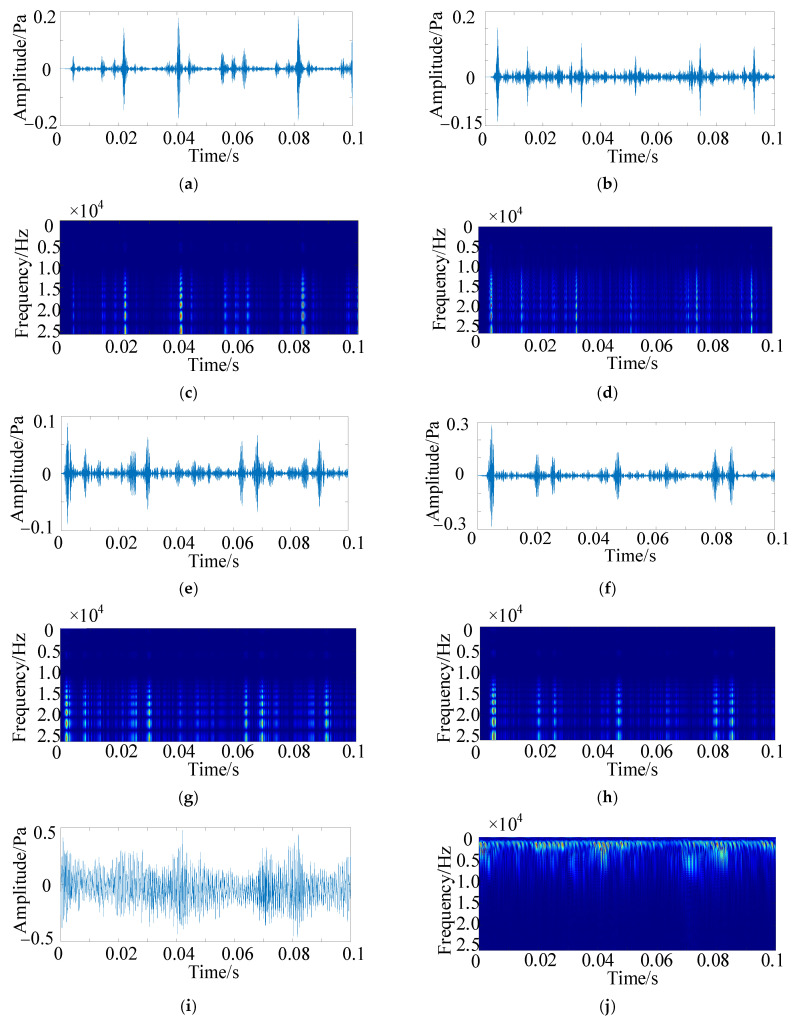
Time domain diagram and time-frequency diagram. (**a**) time-domain diagram of inner raceway fault (0.3 mm); (**b**) time-domain diagram of inner raceway fault (1.0 mm); (**c**) time-frequency diagram of inner raceway fault (0.3 mm); (**d**) time-frequency diagram of inner raceway fault (1.0 mm); (**e**) time-domain diagram of outer raceway fault (0.3 mm); (**f**) time-domain diagram of outer raceway fault (1.0 mm); (**g**) time-frequency diagram of outer raceway fault (0.3 mm); (**h**) time-frequency diagram of outer raceway fault (1.0 mm); (**i**) time-domain diagram under normal conditions; (**j**) time-frequency diagram in normal state.

**Figure 17 sensors-25-05338-f017:**
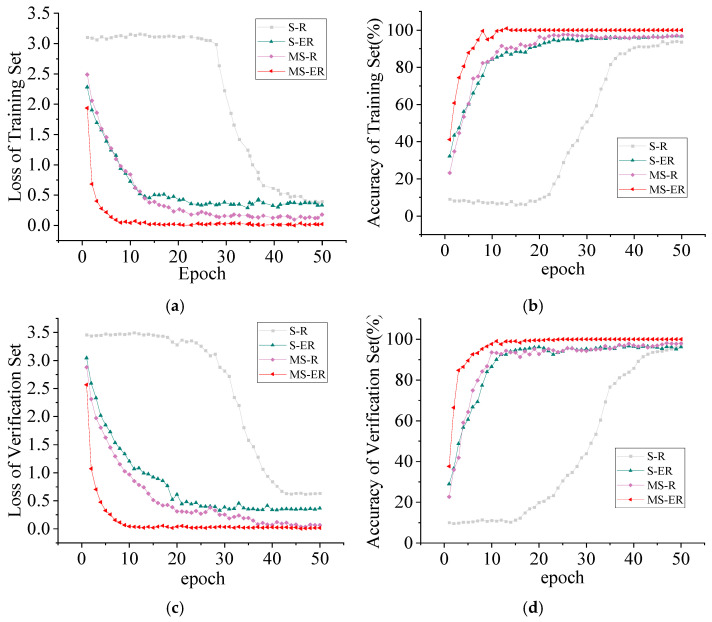
Comparison of ablation experiments. (**a**) Iteration curve of the training set loss function; (**b**) training set accuracy curve; (**c**) validation set loss function iteration curve; (**d**) validation set accuracy curve.

**Figure 18 sensors-25-05338-f018:**
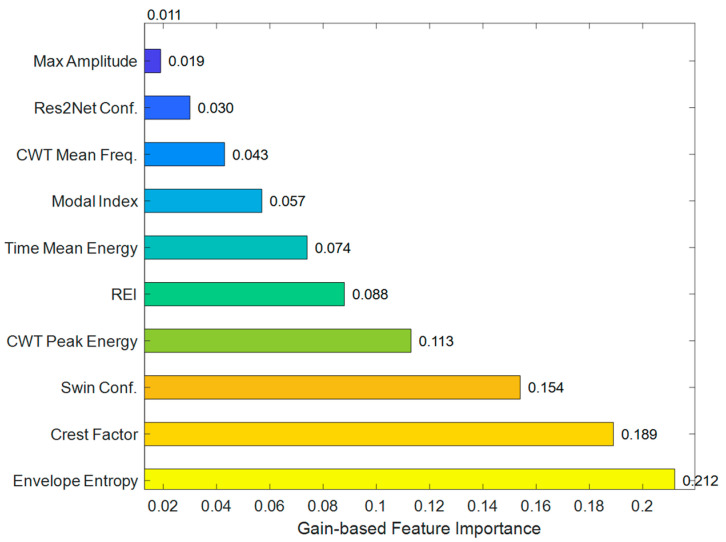
XGBoost feature importance ranking.

**Figure 19 sensors-25-05338-f019:**
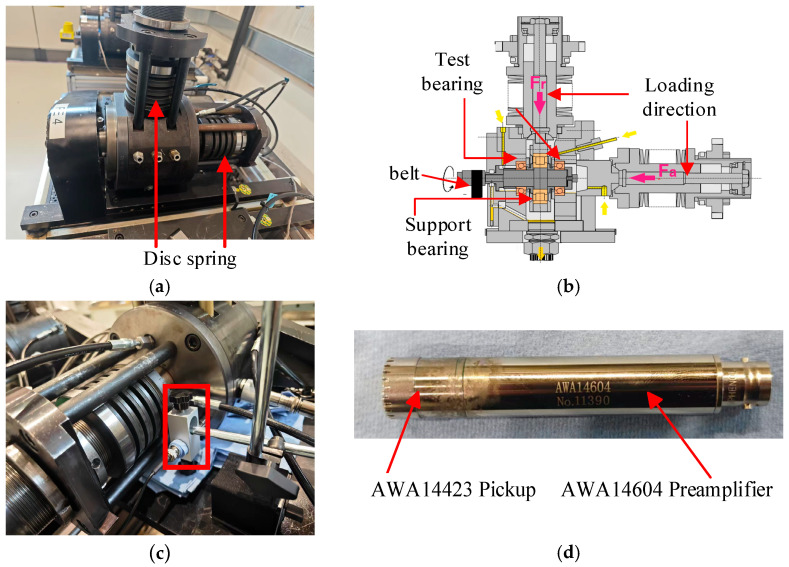
Test instrument. (**a**) Test bench; (**b**) bench structure diagram; (**c**) acoustic sensor installation location; (**d**) acoustic sensor module.

**Figure 20 sensors-25-05338-f020:**
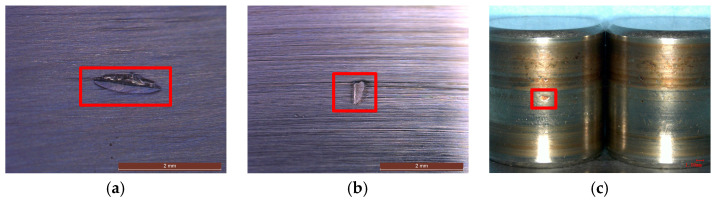
Photos of bearing component fault. (**a**) Inner raceway fault; (**b**) outer raceway fault; (**c**) rolling element fault.

**Figure 21 sensors-25-05338-f021:**
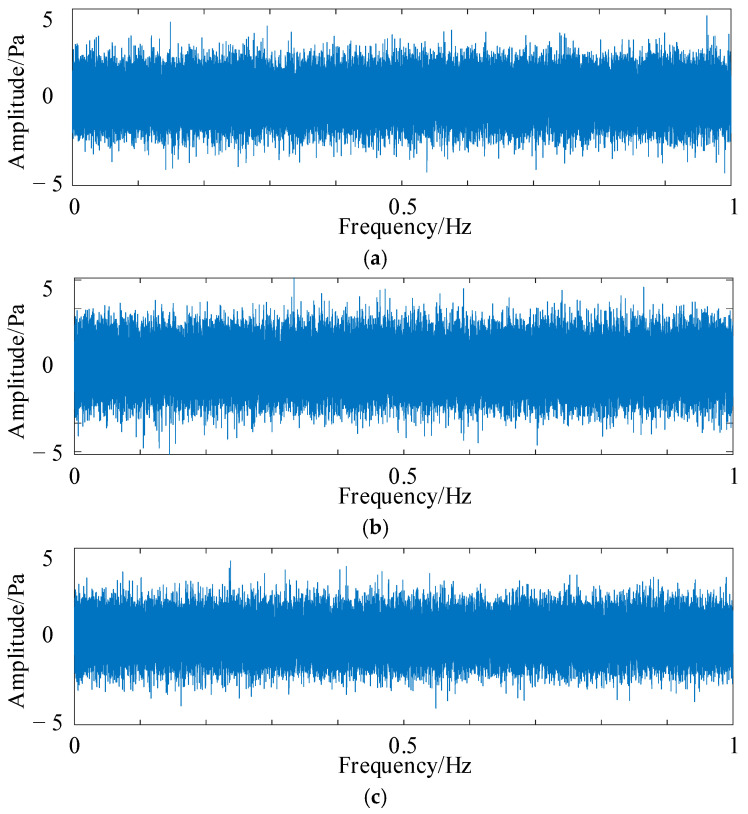
Acoustic signals of different bearing conditions. (**a**) Inner raceway fault; (**b**) outer raceway fault; (**c**) rolling element fault.

**Figure 22 sensors-25-05338-f022:**
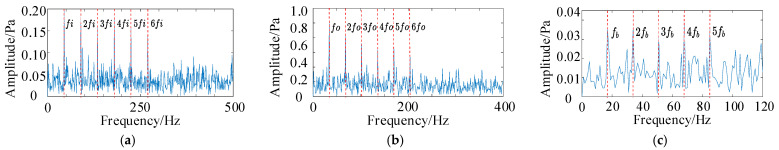
Envelope spectrum of different fault types. (**a**) Envelope spectrum of the inner raceway (IMF4); (**b**) envelope spectrum of the outer raceway (IMF4); (**c**) envelope spectrum of the rolling element (IMF5).

**Figure 23 sensors-25-05338-f023:**
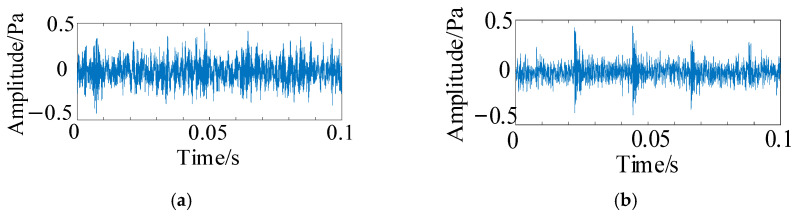
Time-domain diagrams and CWT time-frequency diagrams of different faulty bearings. (**a**) Health status time-domain graph; (**b**) time-domain diagram of inner raceway fault; (**c**) time-frequency diagram of health status; (**d**) time-frequency diagram of inner raceway fault; (**e**) time-domain diagram of outer raceway fault; (**f**) time-domain diagram of rolling element fault; (**g**) time-frequency diagram of inner race fault; (**h**) time-frequency diagram of inner race fault.

**Figure 24 sensors-25-05338-f024:**
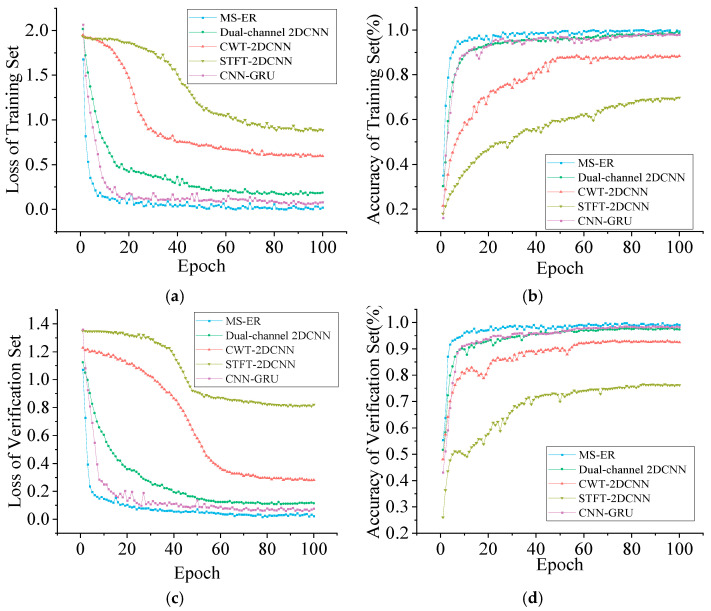
Iteration curves of different models. (**a**) Training set loss function iteration curve; (**b**) training set accuracy curve; (**c**) validation set loss function iteration curve; (**d**) validation set accuracy curve.

**Figure 25 sensors-25-05338-f025:**
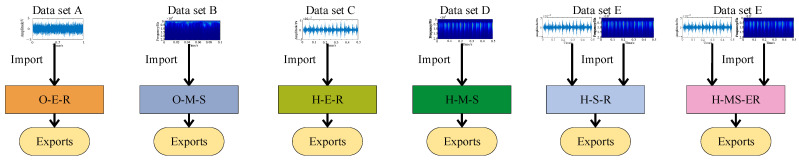
Schematic diagram of different model structures.

**Figure 26 sensors-25-05338-f026:**
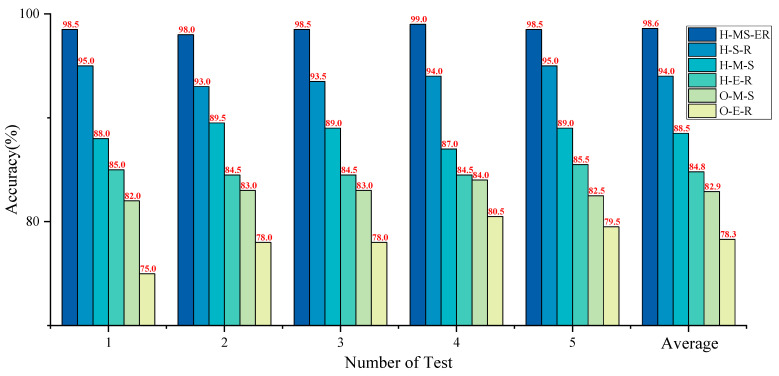
Ablation test results diagram.

**Figure 27 sensors-25-05338-f027:**
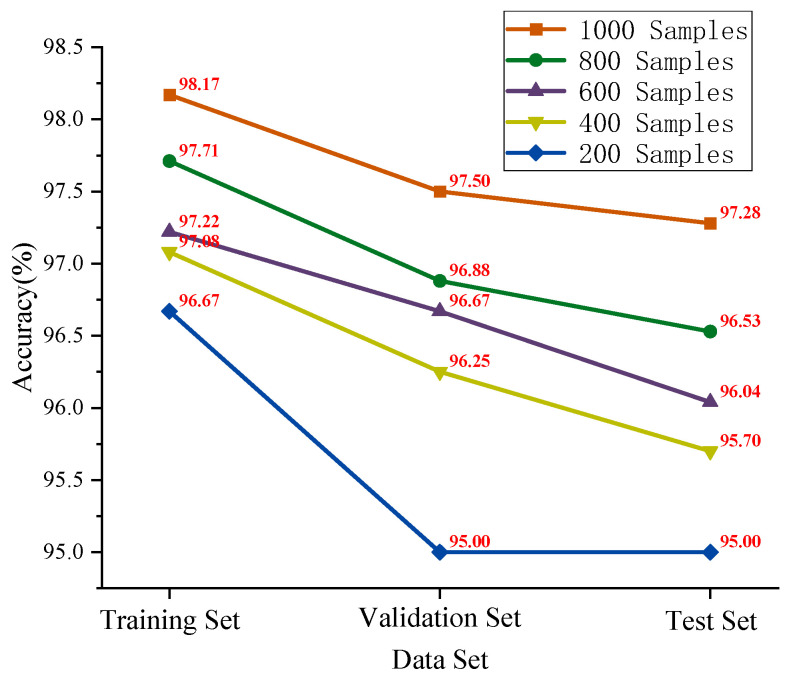
Model accuracy across different samples.

**Figure 28 sensors-25-05338-f028:**
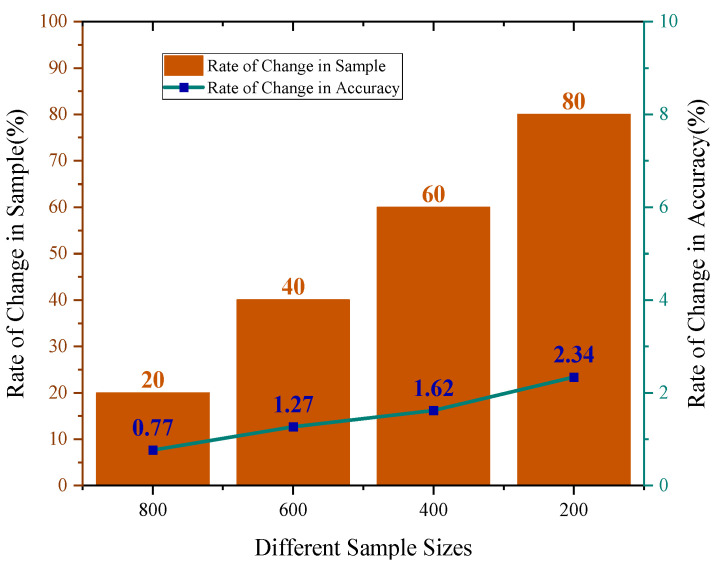
Rate of change in accuracy on the test set.

**Figure 29 sensors-25-05338-f029:**
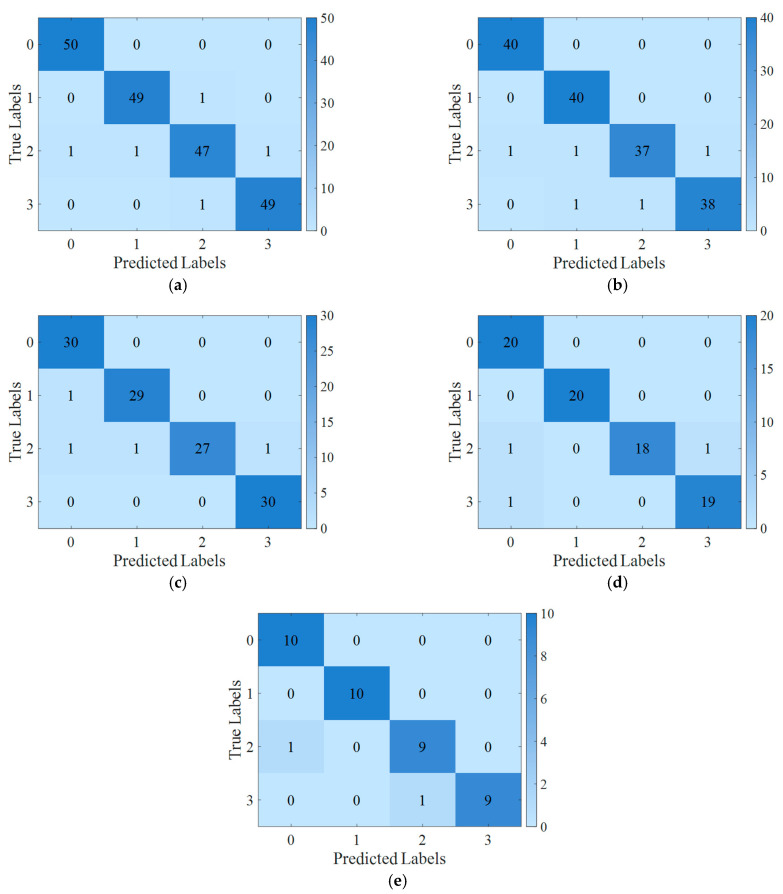
Confusion matrix of different samples. (**a**) Confusion matrix diagram of 1000 samples; (**b**) confusion matrix diagram of 800 samples; (**c**) confusion matrix diagram of 600 samples; (**d**) confusion matrix diagram of 400 samples; (**e**) confusion matrix diagram of 200 samples.

**Table 1 sensors-25-05338-t001:** Parameters of MSCSA-Swin Transformer.

Layer	Type	Input Dimensions	Output Dimensions	Number of Branches	Convolution Kernel	Parameter Settings
Input1	Input layer	H × W × 3	H × W × 3	3	-	Time-frequency domain graph input
Patch Embedding	Embedding Layer	H × W × 3	(H/4) × (W/4) × 96	96	4 × 4	Linear embedding, dividing the image into 4 × 4 small patches
Swin-T Stage1	Swin Transformer Block	(H/4) × (W/4) × 96	(H/4) × (W/4) × 96	96	7 × 7	Contains 2 blocks, window size 7, MLP expansion factor 4
Swin-T Stage2	Swin Transformer Block	(H/4) × (W/4) × 96	(H/8) × (W/8) × 192	192	7 × 7	Containing 2 blocks, window size 7, MLP expansion factor 4, downsampling
Swin-T Stage3	Swin Transformer Block	(H/8) × (W/8) × 192	(H/16) × (W/16) × 384	384	7 × 7	Contains 6 blocks, window size 7, MLP expansion factor 4, downsampling
Swin-T Stage4	Swin Transformer Block	(H/16) × (W/16) × 384	(H/32) × (W/32) × 768	768	7 × 7	Contains 2 blocks, window size 7, MLP expansion factor 4, downsampling
MSCSA Module	Attention Module	(H/32) × (W/32) × 768	(H/32) × (W/32) × 768	768	-	Pool(1,2,4,8) → concat → CSA + Intra-FFN

**Table 2 sensors-25-05338-t002:** ECA-Res2Net.

Layer	Type	Input Dimensions	Output Dimensions	Number of Branches	Convolution Kernel	Parameter Settings
Input2	Input layer	H × W × 1	H × W × 1	1	-	Time-domain image input
Res2Net Stage1	Res2Net Block	H × W × 1	H × (W/2) × 64	64	3 × 3 (stride = 2)	Contains 3 blocks, stride 2, number of branches 64, scale dimension 4
Res2Net Stage2	Res2Net Block	H × (W/2) × 64	H×(W/4) × 128	128	3 × 3 (stride = 2)	Containing 4 blocks, stride 2, number of branches 128, scale dimension 4
Res2Net Stage3	Res2Net Block	H × (W/4) × 128	H × (W/8) × 256	256	3 × 3 (stride = 2)	Consisting of 6 blocks, with a stride of 2, 256 branches, and a scale dimension of 4
Res2Net Stage4	Res2Net Block	H × (W/8) × 256	H × (W/16) × 512	512	3 × 3 (stride = 2)	Containing 3 blocks, stride 2, number of branches 512, scale dimension 4
ECA Module	Attention Module	H × (W/16) × 512	H × (W/16) × 512	512	1D-Conv, kernel size = k	GAP → 1D-Conv → Sigmoid → Scale, local cross-branch interaction

**Table 3 sensors-25-05338-t003:** Feature fusion and classification.

Layer	Type	Input Dimensions	Output Dimensions	Number of Branches	Convolution Kernel	Parameter Settings
Concatenation	Feature fusion layer	H/32 × W/32 × 768 + H × (W/16) × 512	H/32 × W/32 × (768 + 512)	1280	-	Branch splicing
1 × 1 Convolution	Branch compression	H/32 × W/32 × 1280	H/32 × W/32 × 1024	1024	1 × 1	Adjust the number of branches
Global Average Pooling	Pooling layer	H/32 × W/32 × 1024	1 × 1 × 1024	1024	-	Global average pooling
Fully Connected	Fully connected layer	1 × 1 × 1024	1 × 1 × C (C is the number of categories)	C	-	Categorized output

**Table 4 sensors-25-05338-t004:** Data fusion process.

Step	Instructions
1. The original acoustic signal x(t)	The input initial vibration signal.
2. HFMD decomposition (HFOA optimized parameters n, L, K, m)	Optimize HFMD parameters using HFOA to decompose signals into multiple modes.
3. Optimal modal time-domain diagram	Select the optimal time-domain modal signal.
4. CWT transform → optimal modal time-frequency diagram.	Perform CWT on the modal signal to obtain the time-frequency diagram.
5. Res2Net + ECA module	Feature extraction from the optimal modal time-domain diagram.
6. Swin Transformer + MSCSA module	Multi-scale attention feature extraction for optimal modal time-frequency diagrams.
7. Channel concatenation (Concat)	Concatenate the features of the two branches.
8. 1 × 1 convolution for channel dimension reduction	Use 1 × 1 convolution to reduce feature channels and decrease computational complexity.
9. Global Average Pooling (GAP)	Pooling operation to generate global feature vectors.
10. Fully connected layer (classification output)	Perform fault classification output.

**Table 5 sensors-25-05338-t005:** Presents a description of the QU-DMBF data samples.

Fault Type	Fault Diameter	Training Set	Validation Set	Test Set	Tag Value
Normal state	0.00 mm	180	60	60	0
Inner raceway fault	0.35 mm	180	60	60	1
Inner raceway fault	0.40 mm	180	60	60	2
Outer raceway fault	0.35 mm	180	60	60	3
Outer raceway fault	0.40 mm	180	60	60	4
Total	--	900	300	300	--

(In this paper, the CWT time-frequency diagram and the time-domain diagram of a single optimal component are utilized as samples, which are then input into the MSCSA-Swin Transformer and ECA-RES2net models).

**Table 6 sensors-25-05338-t006:** Test set diagnostic results.

Model	QU-DMBF Accuracy (%)	Number of Parameters (M)	FLOPs (G)
S-R	94.83	1.45	0.98
MS-R	97.17	3.12	2.04
S-ER	96.50	2.98	1.89
MS-ER	100.00	3.95	2.80

**Table 7 sensors-25-05338-t007:** Presents a description of the QU-DMBF data samples.

Fault Type	Training Set	Validation Set	Test Set	Tag Value
Normal state	150	50	50	0
Inner raceway fault	150	50	50	1
Inner raceway fault	150	50	50	2
Outer raceway fault	150	50	50	3
Total	600	200	200	--

(In this paper, the CWT time-frequency diagram and the time-domain diagram of a single optimal component are utilized as samples, which are then input into the MSCSA-Swin Transformer and ECA-RES2net models).

**Table 8 sensors-25-05338-t008:** Accuracy rates of different models on test sets.

Model	CWRU Accuracy (%)	Number of Parameters (M)	FLOPs (G)
STFT-2DCNN	81.0	1.10	0.65
CWT-2DCNN	83.1	1.34	0.84
CNN-GRU	92.9	3.22	2.85
Dual channel 2DCNN	93.1	3.75	2.92
BiLSTM-ResNet	93.5	4.20	3.60
Transformer-TELM	94.8	4.90	3.95
MS-ER	98.8	3.95	2.80

**Table 9 sensors-25-05338-t009:** Fault frequencies of bearing FAG NU218.

Fault Characteristic Frequency	Value
$\mathrm{Inner}\;\mathrm{raceway}\;\mathrm{fault}\;\mathrm{frequency}\;f_{i}$, Hz	45
$\mathrm{Outer}\;\mathrm{raceway}\;\mathrm{fault}\;\mathrm{frequency}\;f_{o}$, Hz	34
$\mathrm{Rolling}\;\mathrm{element}\;\mathrm{fault}\;\mathrm{frequency}\;f_{b}$, Hz	17

**Table 10 sensors-25-05338-t010:** REI values of modal components for different fault types.

Type	IMF1	IMF2	IMF3	IMF4	IMF5	IMF6
Inner raceway	0.096	0.086	0.174	0.031	0.207	--
Outer raceway	0.229	0.165	0.139	0.033	--	--
Rolling element	0.213	0.153	0.129	0.074	0.026	0.265

**Table 11 sensors-25-05338-t011:** Description of test data partitioning.

Fault Types	Training Set	Validation Set	Test Set	Label Value
Normal condition	150	50	50	0
Inner raceway fault	150	50	50	1
Outer raceway fault	150	50	50	2
Rolling element fault	150	50	50	3
Total	600	200	200	--

**Table 12 sensors-25-05338-t012:** Accuracy rates of different models on test sets.

Model	CWRU Accuracy (%)	Number of Parameters (M)	FLOPs (G)
STFT-2DCNN	74.2	1.10	0.65
CWT-2DCNN	82.2	1.34	0.84
CNN-GRU	93.5	3.22	2.85
Dual channel 2DCNN	94.2	3.75	2.92
MS-ER	98.4	3.95	2.80

## Data Availability

The data used to support the findings of this study are available from the corresponding author upon request.

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
