# Peer review of "Fault Diagnosis of Rolling Bearings Based on HFMD and Dual-Branch Parallel Network Under Acoustic Signals"

_sensors, 2025, doi:10.3390/s25175338_

Round 1

Reviewer 1 Report

Comments and Suggestions for Authors

To address the issue of reduced fault diagnosis accuracy caused by variations in data quality across different source domains, this manuscript proposes a rolling bearing fault diagnosis method based on Hibert-Fourier Mode Decomposition and a dual-branch parallel network. The results demonstrate that the accuracy of the proposed method performs well on the open-source dataset and actual test data. There are some concerns as follows:

  1. More related works about Hilbert Huang Transform are suggested to be added.
  2. The motivation for combining HFMD and the dual-channel model should be carefully described.
  3. It is recommended that all images be adjusted and enhanced. The dimensions and proportions of certain images are not aligned with the guidelines. A meticulous examination of the font size and dimensions is also recommended.
  4. The shortcomings and future works of the proposed method are suggested to be added to the result section.
Comments on the Quality of English Language

   The representation and readability of this manuscript should be carefully revised.

Author Response

Dear Reviewer,

It is a great honor and pleasure to receive your valuable review and insightful comments on our manuscript entitled " Fault Diagnosis of Rolling Bearings Based on HFMD and Du-al-Branch Parallel Network under Acoustic Signals" Your constructive suggestions have significantly contributed to improving the overall quality and depth of our work, and have also provided us with new directions and objectives for our future research. We are sincerely grateful for your thoughtful guidance.

Throughout the revision process, we have carefully studied and addressed each of your comments with the utmost diligence and respect. We fully recognize that every excellent academic paper benefits greatly from the rigorous scrutiny and insightful feedback of experienced experts like yourself. Your suggestions have helped us optimize the model design, refine the experimental validation, and enhance the clarity and scientific rigor of the paper.

If you have any further comments or suggestions, we would be truly grateful to receive them. We remain committed to addressing any issues to the best of our abilities, in pursuit of the highest academic standards. Under your guidance, we sincerely hope this work will evolve into a more robust and valuable contribution to the field.

Thank you once again for your time, support, and professional insight. We genuinely appreciate the opportunity to revise our work under your careful review.

With deep respect and sincere gratitude,

Reviewer 2 Report

Comments and Suggestions for Authors

Abstract: The abstract better first address the existing issues in rolling bearing fault diagnosis, then explain why traditional methods are insufficient, leading to the proposed improvements made by the authors in this paper.

Introduction: The introduction should provide a literature review of methods based on physical models, signal processing techniques, and data-driven approaches, highlighting their advantages and disadvantages. It is also recommended to compare methods that combine physical models with deep learning, such as "Probabilistic fatigue life prediction in additive manufacturing materials with a physics-informed neural network framework" Expert Systems with Applications. The literature in the introduction is insufficient, and the authors should compare more methods related to domain transfer and Transformer models. Additionally, applications of neural networks, such as "Multi-modal imitation learning for arc detection in complex railway environments" (IEEE Transactions on Instrumentation and Measurement), and "Transformer and Graph Convolution-Based Unsupervised Detection of Machine Anomalous Sound Under Domain Shifts" (IEEE Transactions on Emerging Topics in Computational Intelligence), should also be discussed. The limitations of these methods and areas for improvement should be clearly explained.

Model Innovation: Section 3.2, which discusses the model’s innovations, better be moved to the introduction for readers to quickly understand the contributions.

Methodology: The research process in Section 3.3, which focuses on HFMD and the dual-branch parallel network in bearing fault diagnosis, should also be presented earlier in the methodology section. A pseudocode for the model should be provided, followed by a description of each module. The integration of these modules should also be explained.

Each module (e.g., Sections 4, 5, 11, and 12) should explicitly list the input and output variables for each module.

Excessive Content: Sections 2.4.1 (Res2Net), 2.4.2 (Efficient Channel Attention), and 2.3.1 (Swin Transformer) introduce well-established techniques, and I recommend reducing these sections or removing them altogether.

Experimental Section: The experimental section should compare the parameters and FLOPS of the various models. It also lacks comparisons with SOTA methods from 2024 and 2025, which should be added. Furthermore, the paper should discuss under what circumstances the model performs poorly and provide suggestions for future improvements.

Author Response

(The authors gave the same response as above.)

Reviewer 3 Report

Comments and Suggestions for Authors

To authors:

General recommendation: Major revision is required.

In this study, a rolling bearing fault diagnosis method based on Hilbert-Fourier Mode Decomposition and a dual-branch parallel network has been established for fault diagnosis. The topic of the manuscript is interesting, and the manuscript contains some merits, so it can be reconsidered after resolving the following issues:

1/ Figure 1 is too small. It should be revised in the revision.

2/ From Figure 3, the author decided to use HFOA, the number of iterations should be mentioned in the text.

3/ A comparison between F1, F2, F3, F4 should be more discussed.

4/ Figure 20: how to determine the frequencies fb, 2fb ,… in Figure 20(c)?

5/ It is important to determine the noise based on the experiment test. How did the author examine the based noise in the experiment?

6/ The conclusion section might be simplified. All findings should be given briefly, item by item.

Author Response

(The authors gave the same response as above.)

Reviewer 4 Report

Comments and Suggestions for Authors

The general structure of the manuscript raises some concerns. Some parts of the manuscritp seem only slightly related to each other. For example, the authors use almots four pages to describe the male-to-female ratio of Eagle-Fish, which is not directly used in further computational models. 

Another serious issue is related to the reproducibility of the presented results. It is commonly understandable that the proposed intelligent fault diagnosis algorithms should be compared in order to prove their advatages and drawbacks. A common strategy is using standard and openly accessible datasets. A typical example for fault diagnosis based on vibration signal analysis is the Case Western University Dataset. The authors use acoustical signals - but still the authors should provide space for other researchers to explore and to propose maybe even better algorithms and approaches. 

Back to the acoustic-based fault diagnosis. This may work well in laboratory based environment (what is actually performed in this article) but could fail in real-world industrial environment full of external noises and disturbances. It is well-known that vibration signals are much more rebust to external disturbances. Therefore, the proposed approach is hardly applicable in real-world industrial settings. 

Finally, the general computational setup requires a serious improvement. The authors do not pay sufficient attention in explaining why the proposed schematic structures (layers, blocks, parameters) are optimal for solving the specific diagnostic tasks. 

Author Response

(The authors gave the same response as above.)

Reviewer 5 Report

Comments and Suggestions for Authors

The manuscript addresses a relevant and timely challenge in smart manufacturing—sensor fault diagnosis using information fusion and machine learning. The topic is well aligned with current trends, and the overall structure of the paper is clear. However, in its current form, the manuscript requires several important revisions before it can be considered for publication.

– The main contribution lies in the combination of feature-level and decision-level fusion, but the technical description of how features are extracted and combined is insufficient. Are the fusion weights predefined, learned through training, or assigned empirically? I would recommend including a diagram or flowchart to clearly illustrate the data fusion process.

– While the use of XGBoost is mentioned, it is not clear how hyperparameters were tuned (e.g., manually, via grid search, cross-validation). Furthermore, there is no explanation of how overfitting was controlled—this is especially relevant when working with fused features.

– The description of the datasets used is vague. Are these datasets publicly available or collected in-house? What is the number of samples, class distribution, and train/test split? I recommend providing a table summarizing key dataset characteristics.

– Although the results of the proposed method are presented, there is no comparison with alternative approaches, such as single-sensor classifiers or classical signal processing methods. Including benchmark models would be crucial to validate the effectiveness of the proposed approach.

– The decision-level fusion appears to rely on majority voting, which is simple but lacks transparency. I recommend presenting feature importance measures (e.g., XGBoost importance rankings or SHAP plots). Additionally, a discussion of the limitations of the method and its potential applicability in real industrial environments would strengthen the manuscript.

Author Response

(The authors gave the same response as above.)

Round 2

Reviewer 1 Report

Comments and Suggestions for Authors

All my concerns have been addressed.

Reviewer 2 Report

Comments and Suggestions for Authors

  Thank you for carefully addressing the feedback provided in my previous review. I have thoroughly reviewed your revised manuscript and am pleased with the improvements you have made. Your thoughtful revisions have enhanced the quality and clarity of the research presented.

   As a final suggestion, I recommend that you carefully check the grammar, punctuation, and formatting of the manuscript to ensure it fully complies with the Sensor's requirements before submitting the final version. This will help to further improve the professionalism and readability of your work.

Reviewer 3 Report

Comments and Suggestions for Authors

The paper can be accepted in the present form.

Reviewer 4 Report

Comments and Suggestions for Authors

The authors did perform a relevant revision. The reviser manuscript is recommended to be accepted in the Journal. 

Reviewer 5 Report

Comments and Suggestions for Authors

After the authors have responded to my questions, I no longer have any major concerns regarding the content of the manuscript.